# Fluorescence imaging of individual ions and molecules in pressurized noble gases for barium tagging in $^{136}$Xe

N. K. Byrnes[1], E. Dey[1], F. W. Foss[2], B. J. P. Jones [1]✉, R. Madigan[2], A. D. McDonald[1], R. L. Miller [2], L. R. Norman[1], K. E. Navarro[1], D. R. Nygren[1] & NEXT Collaboration*

The imaging of individual Ba$^{2+}$ ions in high pressure xenon gas is one possible way to attain background-free sensitivity to neutrinoless double beta decay and hence establish the Majorana nature of the neutrino. In this paper we demonstrate selective single Ba$^{2+}$ ion imaging inside a high-pressure xenon gas environment. Ba$^{2+}$ ions chelated with molecular chemosensors are resolved at the gas-solid interface using a diffraction-limited imaging system with scan area of $1 \times 1$ cm$^2$ located inside 10 bar of xenon gas. This form of microscopy represents key ingredient in the development of barium tagging for neutrinoless double beta decay searches in $^{136}$Xe. This also provides a new tool for studying the photophysics of fluorescent molecules and chemosensors at the solid-gas interface to enable bottom-up design of catalysts and sensors.

Single-molecule fluorescence imaging (SMFI) is a Nobel Prize-winning technique[1] that has enabled major advances in biochemistry and cellular imaging[2]. SMFI enables super-resolution microscopy[3], illuminating features in cells far below the diffraction limit. In addition to transformationally advancing the resolution of optical microscopes, SMFI also represents the ultimate frontier in analytic chemistry. By custom chemosensor design, single molecules of specific analytes can be sensed both in vitro and in vivo[4]. SMFI implemented at the gas-solid interface has the potential to open a host of new applications, though, beyond state-of-the-art microscopy methods, molecular and supramolecular synthesis approaches are required for its realization. An especially compelling application that is currently driving the development of SMFI at the gas-solid interface is barium tagging[5,6], the sensing of individual ions of Ba$^{2+}$ in xenon, which could significantly increase the discovery reach of neutrinoless double beta decay ($0\nu\beta\beta$) searches. The identification of individual ions at a high-pressure gas interface is an important advance for both SMFI and $0\nu\beta\beta$.

A profound open question in physics today is whether the neutrino is its own antiparticle (a Majorana fermion). The only known sensitive way to establish the Majorana nature of the neutrino is via direct observation of $0\nu\beta\beta$[7,8]. In this hypothetical radioactive process, two neutrons (or protons) in a nucleus transform into two electrons (positrons) with the emission of no neutrinos or anti-neutrinos. This process violates the Lepton number[9], would guarantee the generation of at least a small Majorana neutrino mass through loop corrections[10], and could provide a compelling window into the mechanism leading to the dominance of matter over antimatter in the Universe[11]. Its detection would provide a revolutionary insight into the nature of neutrino mass, potentially the only directly observed manifestation of physics above the electroweak scale.

One isotope that has been used in many $0\nu\beta\beta$ searches to date is $^{136}$Xe, which can decay to $^{136}$Ba via nuclear decay $^{136}_{54}$Xe$\rightarrow^{136}_{56}$Ba$+2e^-$. Because the daughter nucleus in the final state is very heavy relative to the electrons, they carry away almost all of the available energy, producing a nearly mono-energetic line at the Q-value for the decay, $Q_{\beta\beta} = 2457.8$ keV. Reconstructing electron energy deposits in media enriched in $^{136}$Xe has enabled searches for this process, with the current strongest limit being $2.3 \times 10^{26}$yr at 90% confidence level[12].

The expected rate of $0\nu\beta\beta$ depends on the neutrino masses and mixing parameters, nuclear matrix elements[13], and phase space factors[14]. Neutrino oscillations have measured two characteristic mass-squared differences between neutrino states, $\Delta m_{32}^2 \sim 2.4 \times 10^{-3}$eV$^2$ and $\Delta m_{21}^2 \sim 7.4 \times 10^{-5}$eV$^2$ [15]. The three neutrino masses are thus either organized with a large gap between the heavier pair (normal ordering) or the lightest pair (inverted ordering). If the value of the lightest neutrino mass is smaller than 100 meV, the expected range of lifetimes for

[1]Department of Physics, University of Texas at Arlington, Arlington, TX, USA. [2]Department of Chemistry and Biochemistry, University of Texas at Arlington, Arlington, TX, USA. *A list of authors and their affiliations appears at the end of the paper. ✉e-mail: ben.jones@uta.edu

$0\nu\beta\beta$ depends strongly on this ordering. In the inverted case, or if the lightest neutrino mass is greater than 100 meV under the normal ordering, $0\nu\beta\beta$ ought to be discoverable with a lifetime $\tau \leq 10^{28}$ yr. To achieve such sensitivities, detectors with $\geq 1$ Ton of the double beta decay isotope and background levels (measured in counts per ton-year in the energy region of interest (ROI)[16]) of order $b < 0.1$ ct (ton yr ROI)$^{-1}$ are required[17]. Factors between 20 and 2000 beyond existing demonstrated technologies are required to meet these goals for the immediately forthcoming generation of experiments. Beyond this ton-scale phase[18–21], the task of extending sensitivities into the non-degenerate, normal mass ordering band of parameter space appears truly formidable. The relevant experiments would employ far larger quantities of active isotope[22] and require still lower background indices. A distinct but related question is how to confirm a signal of $0\nu\beta\beta$ following a suggestive but inconclusive hint from the coming generation of still-background-limited experiments. New and possibly radically new ultra-low background technologies will be required to meet these challenges.

Two neutrino double beta decays ($2\nu\beta\beta$) are one source of background to $0\nu\beta\beta$ and can be efficiently rejected by technologies with full-width-half-maximum (FWHM) energy resolution $E_{FWHM} \leq 1\%$. The remaining backgrounds in contemporary experiments originate from radiogenic gamma rays in detector materials, cosmogenic material activation, and in principle, solar neutrino interactions in the detector. One especially promising technical approach to remove all such events and hence reach the ultra-low background limit is barium tagging—identification of the $^{136}$Ba daughter ion produced in the double beta decay of $^{136}$Xe[5]. An efficient and selective barium ion tag could reduce contamination from all backgrounds except for $2\nu\beta\beta$ to effectively zero. Demonstration of a method of capture and imaging of barium ions from one to several tons of xenon requires significant advances in instrumentation. To provide either a significant sensitivity boost or a signal confirmation, barium ions must be captured and then identified with high efficiency (greater than around 50% to avoid substantial loss of sensitivity relative to current analysis methods[23]) in coincidence with electron energy deposits near the Q-value reconstructed with resolution better than 1% FWHM to reject the two-neutrino mode. The two-neutrino decay mode also produces barium ions at a rate of around 5 per kilogram per day[24], which imposes a loose spatio-temporal requirement on the coincidences that must be established between the ion and electron signatures. 3D imaging of electron tracks may provide further confirmation of the two-electron topology[25,26], in principle enabling a robust three-fold coincident signature.

Much progress has been made on promising methods for single barium ion or atom identification in liquid and gaseous xenon[27–41]. Most approaches to barium tagging apply a form of fluorescence imaging, exciting transitions in atoms, molecules, or materials that are caused by the presence of a barium atom or ion. In xenon gas experiments, the target charge state for barium tagging is the dication $Ba^{2+}$ due to the absence of recombination from thermalized electrons around the decay[42]. This ion has no low-lying atomic fluorescence lines to access with visible lasers. However, when chelated within a suitable organic molecule, its appearance can, in principle, be observed via single-molecule fluorescence imaging (SMFI). The proposal to use SMFI to tag the daughter ion in $0\nu\beta\beta$ was first outlined in refs. 6,34. Commercial dyes developed for $Ca^{2+}$ sensing in biological applications were demonstrated as sensitive barium tagging agents in the solution phase. Soon thereafter, single ion sensitivity was achieved[32]. The model system in that work, liquid droplets within a polymer matrix, is not well representative of conditions within a xenon gas time projection chamber. Several aspects of the chemistry of binding and fluorescence of the commercial dyes were found to be inadequate for dry barium sensing, instigating a program of novel organic fluorophore development, which has culminated in multiple candidates for single $Ba^{2+}$ sensing based on crown-ether derivatives[33,36,39,43–46]. These molecules have been used to demonstrate single-molecule sensitivity to barium in dry and solvent-less conditions[35,46] and shown via electron microscopy to react with neutral barium perchlorate in vacuum[39]. An active ongoing R&D program is underway within the NEXT collaboration to bring these techniques to fruition in barium tagging sensors for a future ton- to multi-ton neutrinoless double beta decay experiment.

Experiments searching for $0\nu\beta\beta$ in $^{136}$Xe typically employ time projection chambers filled with purified, pressurized, or liquified xenon at part-per-billion purity in oxygen and water to avoid electron attachment. To augment such a detector with a barium tagging system, two distinct problems must be overcome: first, how to bring the ion to a sensor[37,38] or sensor to an ion[33]; and second, how to sense the arrival of $Ba^{2+}$ with single-ion precision within a large, pure volume of xenon. This is a complex environment in which to realize single-ion microscopy, with no commercial devices or past proofs-of-principle available. This paper presents a demonstration of single $Ba^{2+}$ imaging within the working medium of a time projection chamber.

Beyond $0\nu\beta\beta$, the development of single-molecule fluorescence imaging technologies at gas-solid interfaces has far-reaching implications for the bottom-up design of catalytic and sensing materials[47–49]. The heterogeneous catalysis of gases is critical to advanced energy technologies and sustainable materials[50]. Likewise, gas-phase sensors are critical to the monitoring of industrial, defense, and environmental processes[51–53]. SMFI in these systems enables a deeper understanding of kinetics and thermodynamics that are dependent on local structures and orientations within functional materials[47]. While the field of catalysis has seen recent growth in SMFI to observe shape and orientation at the single metal atom scale, similar advancements are relatively underdeveloped in gas analyte sensing. The instrumentation, materials, and approach to SMFI developed in our work open the door to researchers in both catalysis and gas sensing to observe single molecule binding and surface dynamics. The range of pressures that the microscope functions under enables catalysis and sensor study at real-system operational pressures.

A detailed description of the instrument developed for this work is provided in the methods section. Briefly, a fluorescence microscope based on nanometer precision vacuum stages is mounted on the inside of a high-pressure chamber connected to a recirculating xenon gas system. Images are recorded by an electron-multiplying CCD camera following laser excitation of barium-induced fluorescence in $Ba^{2+}$ selective organic chemosensors. The system has been characterized using emission from BODIPY fluorophores[54] excited at 488 nm. The configuration for this characterization study is shown in Fig. 1.

## Results

A raw data image of sparsely distributed molecules, produced by rastering in 33 μm steps over a 1 mm$^2$ surface area and auto-focusing on the single molecule candidates at each point is shown in Fig. 2. The unprocessed raw data shows the effects of non-uniform illumination over the laser field of view, with each point in the raster-scan having a bright center and dimmer tails, as well as some bright spots where crystals of fluorophore have settled, among the field of isolated single molecules. Some streak-like features from the motion of the solution during spin coating are also visible. Nevertheless, the capability for single-molecule imaging over this large surface area is clearly demonstrated, with resolution near the Abbe diffraction limit. Since the exposure at each raster point is 500 ms, a scan of this size is performed in approximately ten minutes. Further studies of the system imaging resolution can be found in the methods section.

To demonstrate single $Ba^{2+}$ ion imaging in high-pressure gas, slides were coated with nanomolar concentrations of the Ion Potassium Green (IPG-1) fluorophore reported in ref. 46. Solutions were prepared at $10^{-8}$ M concentration, with and without barium ions supplied via 100 mM barium perchlorate in water, and activity was compared between scan surfaces coated with $Ba^{2+}$ chelated and unchelated solutions. The optical system for this test was configured with a 510 nm

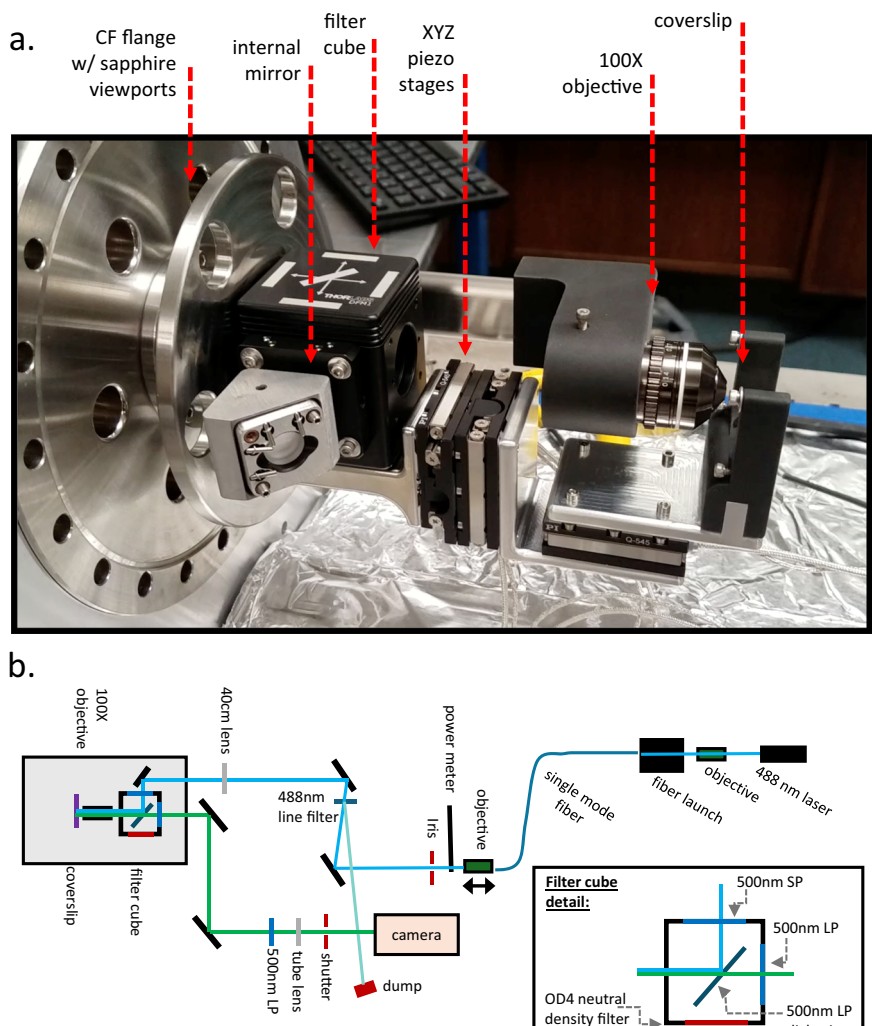

**Fig. 1 | Picture of the apparatus developed for this study. a** The head of the high-pressure microscope that sits inside the pressure chamber. **b** Optical paths and components in the high-pressure single-molecule microscope. SP and LP refer to short-pass and long-pass optical filters, respectively.

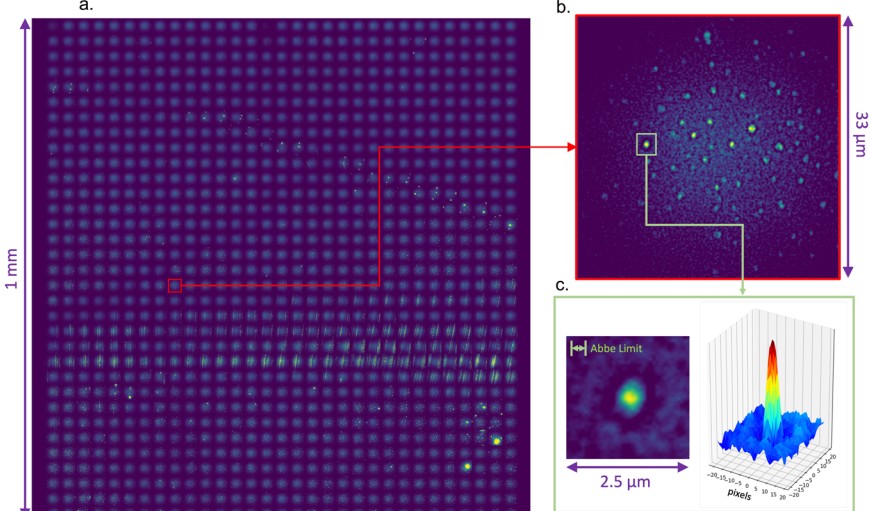

**Fig. 2 | Large-scale raw data image of BODIPY molecules drip-coated onto a slide surface. a** Full 1 mm × 1 mm image; (**b**) zoom into one raster frame of 33 μm × 33 μm; (**c**) resolved single molecule in a 2.5 μm × 2.5 μm square. The image is resolved with a point-spread function close to the Abbe Diffraction Limit.

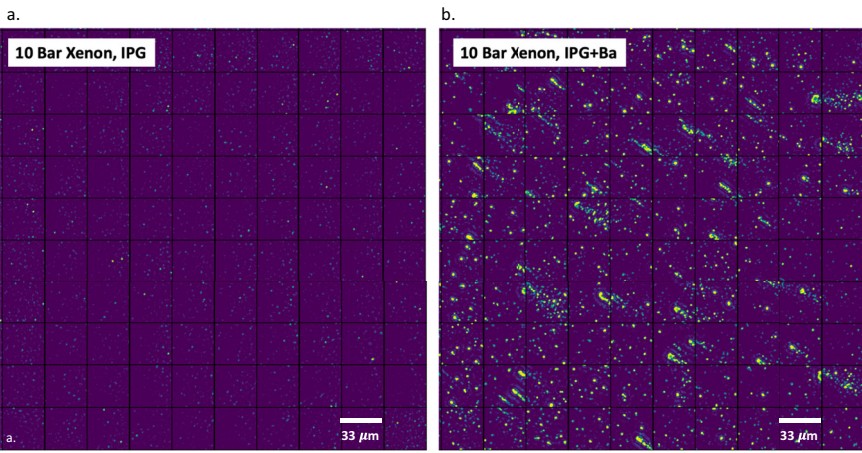

**Fig. 3 | Effect of Ba²⁺ addition to the sensor system.** Images show a slide spin-coated in IPG-1 chemosensor, showing the activity with (**a**) and without (**b**) added Ba²⁺.

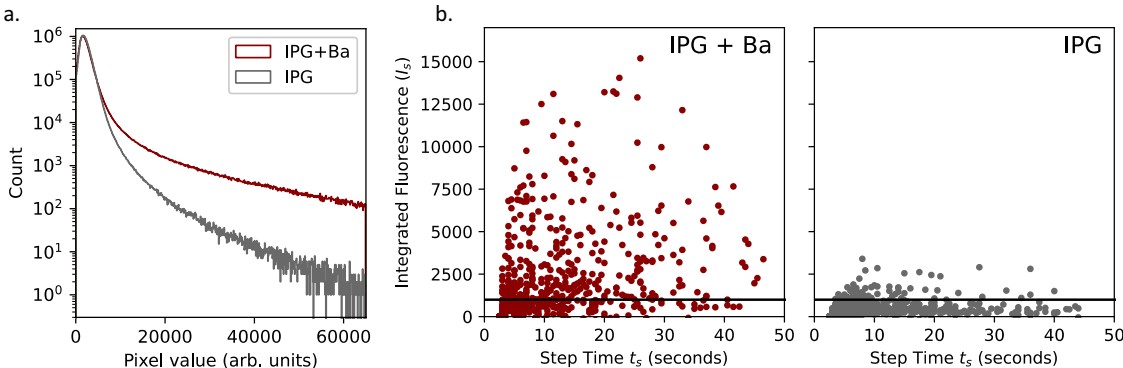

**Fig. 4 | Statistical response of chemosensors to the addition of Ba²⁺ in xenon gas. a** Pixel intensity distribution for images taken using IPG vs IPG + Ba in 10 bar xenon gas. **b** reconstructed single molecule candidate brightnesses and step times for IPG vs IPG + Ba in 10 bar xenon gas. The plotted points are obtained over seven exposure regions on a single slide, and the trend is found to be repeatable over multiple slides. The horizontal line shows a cut that we place on this distribution for selecting single-ion candidate spots.

laser with 10 mW incoming laser power and a filter set suitable for this dye, as described in the methods section. The vessel was then sealed, evacuated, and filled with cleaned, pressurized xenon. Figure 3 shows images taken under purified 10 bar xenon gas for slides prepared with and without added Ba²⁺. In this image each frame has been Fourier transformed and spatially filtered. A low pass filter is used to remove the broad background from residual glass fluorescence, and a high pass filter to remove speckles associated with thermal CCD noise. The resultant images display clear bright spots consistent with single molecule emission.

A clear increase in activity is observed upon the addition of Ba²⁺. Some visible streaks in the distribution of single molecule candidates emanate in a radial direction and are a result of the spin-coating protocol, where the solution from which molecules are deposited leaves some residual droplets as it dries. Single molecules remain resolvable within the streaks, though our analysis methods preferentially select those that are isolated single spots. The raw pixel histogram indicates a dramatic increase in activity in bright pixels, as shown in, Fig. 4, left. A small number of very weak emitters are present in the Ba²⁺-free runs, though the bright spots associated with Ba²⁺-bound IPG molecules are unambiguously identified as being present only in the Ba²⁺-chelated runs.

As a control experiment, blank coverslips with no fluorophore deposited were also scanned. The characteristic diffuse glow from background fluorescence in the slide was observed in all locations. However, no localized bright emitters could be brought into focus at

any point on the slides. This suggests that the background due to accidental fluorescent molecules in the environment is extremely low, and has proven to be unquantifiably so in the present system. Because no emitters could be found, the microscope could not be focused and so no images from the blank slide surfaces can be reported.

The hallmark of single-molecule fluorescent imaging is bright points that are characterized by their sharp photobleaching transitions. A set of analysis tools for single-molecule fluorescence imaging and analysis have been developed and are applied to the images captured by the high-pressure microscope. To produce a time sequence, the first ten images in a sequence are summed, and a list of local fluorescent maxima ordered by their intensity is used to determine initial points of interest. Each frame is Fourier transformed and convolved with a bipolar filter kernel consisting of one normalized narrow Gaussian ($\pm 4$ pixels) with positive amplitude and one normalized broad Gaussian ($\pm 8$ pixels) with negative amplitude. This filter serves to subtract the local background while integrating fluorescence within the characteristic width of the fluorescent emission area. A cut on the maximal identified step confidence metric (described in the "Methods" section) in each trace is used to identify single molecule photobleaching event candidates.

Such candidates are identifiable in all conditions tested, including ambient air, vacuum, and pressurized argon and xenon, in Ba²⁺ spiked runs. The time and intensity distributions of the reconstructed steps in tests with and without Ba²⁺ are provided in Fig. 4, right. In these plots, the $y$-axis corresponds to the fluorescence integral up to the

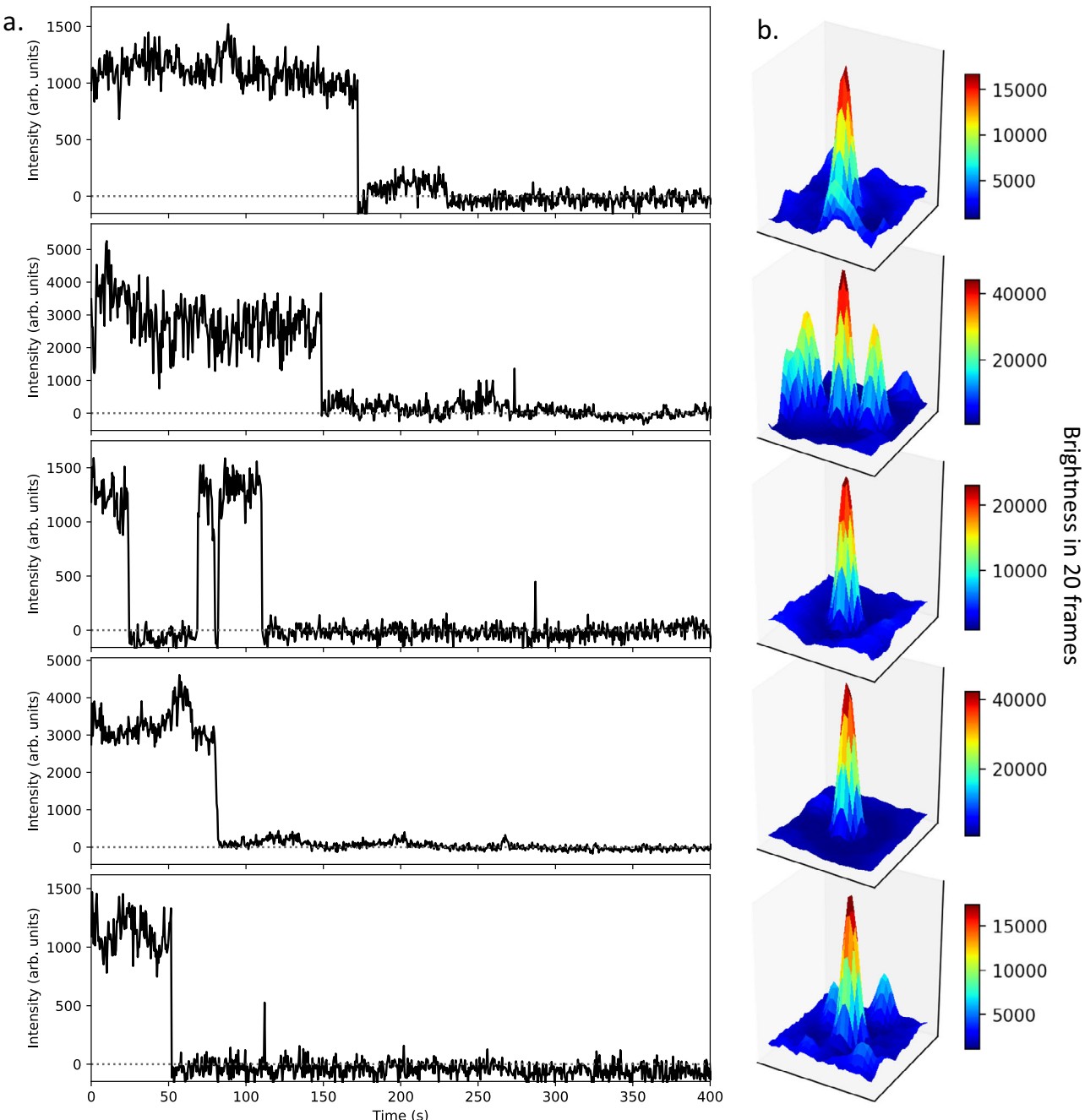

**Fig. 5 | Single barium ions chelated with IPG-1 turn-on chemosensor imaged in 10 bar xenon gas. a** time trace of fluorescence for the identified emitters, with discrete photobleaching and photoblinking steps indicative of single molecule origin. The intensity is taken from the central pixel after applying a double-Gaussian filter as described in the text and thus represents the baseline-subtracted integral over the diffraction-limited fluorescent spot. **b** 2D spatial map of fluorescence around each emitter, integrated for 20 frames before photo-bleaching, showing a well-localized peak in each case. In the case of the second and fourth figures, nearby peaks from adjacent ions are also visible in the 3D image histograms.

photobleaching transition divided by the total time to the step, providing a measure of the average brightness of the emitter. The x-axis corresponds to the step time, which is defined as the time when the most significant change in fluorescence intensity between five pre-samples and five post-samples is observed. Only very small steps are observed in the $Ba^{2+}$-free runs, whereas the $Ba^{2+}$-enriched ones contain large steps associated with single barium ion candidates.

Figure 5, left shows the time traces for some single $Ba^{2+}$ ions identified via single-step photo-bleaching 10 bar xenon gas. To the right of each trace are shown the fluorescence intensity maps corresponding to each, representing the activity recorded on a small subset

of the CCD pixels. Each surface plot shows the integral of the activity between the last 20 frames before the photo-bleaching step. The characteristic photo-bleaching and, in some cases, photo-blinking behavior associated with single-molecule fluorescence imaging is clearly observed in each time series.

Studying the photo-bleaching time distribution illuminates a striking difference in the bleaching dynamics of $Ba^{2+}$-chelated IPG dyes in noble gas vs air environments, with substantially faster bleaching behavior observed in air. The quantitative photo-bleaching lifetime depends on both the local laser intensity and the step-identification protocol, in particular how the latter accounts for multi-step

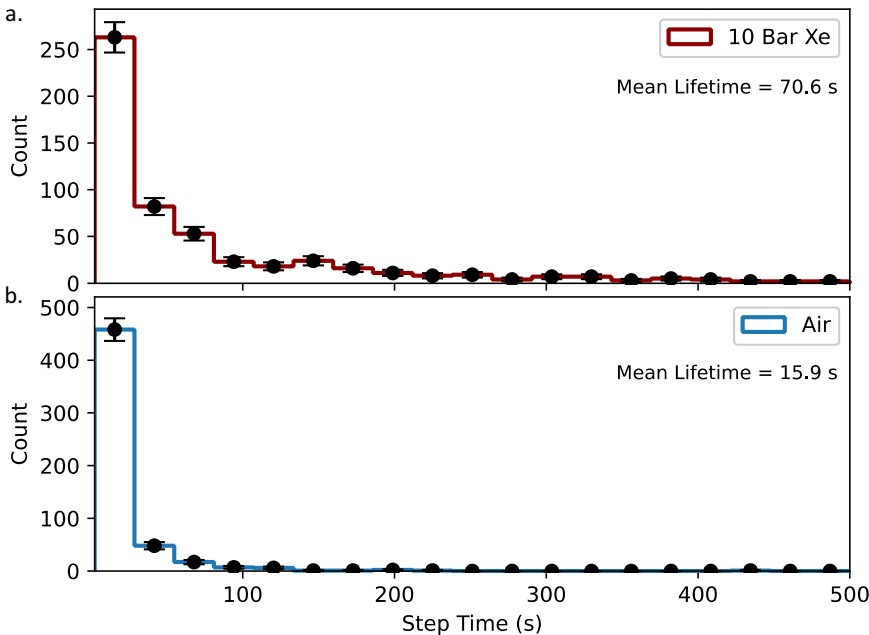

**Fig. 6 | Photo-bleaching time distribution for Ba²⁺ candidates chelated in IPG dyes in xenon and air. a** in xenon gas; (**b**) in air. The *X*-axis is shared between the top and bottom histogram. Error bars indicate representative 1σ statistical errors calculated as $\sqrt{N}$ of detected counts $N$.

trajectories associated with photo-bleaching and photo-blinking are handled. Using the algorithm outlined in the Methods Section, which identifies the first significant (5σ in step confidence) transition in selected each time sequence, the photo-bleaching lifetime in 10 bar xenon gas is extracted to be 70.6 s, whereas in air it is much faster at 15.9 s, as shown in Fig. 6. These values are extracted from accumulated barium ion candidates identified over seven long timescale runs of 500 s each. This observation supports the hypothesis that has been discussed in association with solution-based studies (for example, refs. 55,56), that reactions with oxygen are likely primarily responsible for the bleaching process. Nevertheless, even absent ambient oxygen some photo-bleaching mechanism appears to be present, albeit at a far slower rate. While large datasets of IPG + Ba²⁺ images were not collected in vacuum or argon conditions for this work, visual inspection of the photo-bleaching behavior while imaging in those conditions showed a time profile far more similar to the xenon data than to air, as may be expected under the oxygen-mediated hypothesis.

## Discussion

We have demonstrated a diffraction-limited, high-pressure fluorescence imaging system that is capable of single ion identification in high-pressure gases at the gas-solid interface. Single fluorescent molecules are resolved over large surface areas, and we have demonstrated a sweep over 1 × 1 mm² via a 2D raster scan. An autofocus algorithm that reliably brings emitters as weak as a single molecule into focus is used to provide a map of the focal plane that can be extrapolated for rapid and repeatable imaging. The total possible scan region is in excess of 1 × 1 cm², with effective focusing possible over the full area.

Single Ba²⁺ ions have been imaged in high-pressure xenon gas using turn-on fluorophores, representing a demonstration of imaging of individual Ba²⁺ ions within a candidate active medium of a time projection chamber for 0νββ. Given the large observed binding constants of the crown-ether-based dyes[46], this mode of microscopy appears well suited to providing an efficient barium imaging technique for future xenon 0νββ experiments.

Since barium tagging rejects all radiogenic background events, the radio-purity requirements of the molecular monolayer and barium imaging system are expected to be modest, and likely already to be

met by the current system. Furthermore, if coupled with an ion transport device such as a radiofrequency carpet[38], the demonstrated scan area will also suffice for a realistic barium tagging sensor for a 0νββ experiment. Some modifications are still required for implementation within a time projection chamber, which we now briefly discuss. First, in this work, rastering was performed by moving a slide past a fixed objective lens, due to volume limitations within the vessel. In the final device it should be the converse, with rastering of the objective across a fixed imaging region. Since the image rays are brought out of the vessel in infinity space, this will add little in the way of complexity but require some mechanical adjustments. Second, continuous and lossless operation at this resolution generates a tremendous amount of data (15 GB uncompressed, for the single raster image in Fig. 2). A rational zero suppression algorithm and an online trigger for frames of interest will be advantageous for real-time application, with factors of over 1000 in data reduction to be reasonably expected. These issues require attention but appear manageable. The molecular layer used here for Ba²⁺ sensing is produced by spin coating a sparse group of molecules from the solution; ultimately a fully sensitive Ba²⁺ tagging layer must use densely packed fluorophores. Detailed investigations into the behavior of densely packed fluorophore layers on surfaces[39] and into self-assembled monolayer growth[44], as well as studies of the efficiency of barium ion capture at these surfaces using Ba²⁺ ion beams[57] are currently underway within the NEXT collaboration and represent the next crucial frontier in chemical Ba²⁺ ion sensor development. While important work must be done to realize the ultimate sensing layers, an optical system like the one described in this paper will be capable of imaging ions arriving at their surfaces in a future xenon gas barium tagging detector.

Alongside the ongoing development of molecular synthesis[33,36,39,43–46], ion transport[38] and detector readout modalities[58–61] that can enable barium tagging at the cathode, the new technology demonstrated in this paper represents an important step toward a large, barium tagging xenon gas detector. Such a device holds great promise as a concept for a truly background-free, ton-to-multi-ton scale neutrinoless double beta decay experiment in ¹³⁶Xe. It also represents a step into a new field of high-precision, single-molecule photo-physical fluorescence analysis at the gas-solid interface.

## Methods

To obtain high quality images, any fluorescence microscope must have carefully arranged optical paths for both illumination and image collection. We begin by describing these optical elements in Sec. IV A and IV B, respectively, and then describe the mechanical construction of the device in Sec. IV E. Section IV C describes the alignment and focusing protocol and quantifies some of the imaging metrics of the system.

### Excitation

Past work on single $Ba^{2+}$ imaging for NEXT has shown that single ion candidates can be observed with as little as 0.2 mW of laser power in the field of view (FOV), corresponding to around $1\,W\,cm^{-2}$[32], given molecules with sufficiently high quantum efficiencies. On the other hand, fluorescent molecules that are exposed to large integrated light intensities undergo destructive photobleaching reactions[62], limiting the practically usable laser power for prolonged observation. As such, we have optimized the current system to provide few-mW levels of excitation to enable clear single molecule resolution while offering suitably long (few seconds) observation times to establish the presence of an individual ion. We have also opted for an epi-fluorescence configuration. While through-objective total internal reflection fluorescence (TIRF) imaging was used in our earliest studies of fluorescent single barium ion complexes in order to suppress deeper backgrounds arising in thick samples[32], more recent work demonstrated that for sufficiently thin fluorescent materials, single molecule resolution can also be obtained in an epi-fluorescence mode[35]. The latter approach is more straightforward to realize with remote illumination and forms the basis of our illumination scheme for this device. For this form of microscopy, light is focused on the axis on the back-focal plane of the objective by an external lens, leading to the parallel and uniform passage of light through the focal plane, which is aligned to the surface supporting the fluorescent emitters to be imaged.

We use a series of solid-state lasers as excitation sources, each suitable for a different set of fluorescent dyes. For the characterizations of the optical system performance, we have used a 488 nm laser with power controlled by an adjustable DC power supply. This wavelength choice is sufficiently long to escape the tail of fluorescence from glass that compromises single-molecule imaging with shorter wavelength excitation (at 450 nm and below, the autofluorescence from the objective and substrate proved prohibitive for single-molecule imaging), and is well matched to the absorption peak of the BODIPY[54] molecule used for our optical system characterizations. For $Ba^{2+}$ sensing, a 510 nm laser is used to excite the IPG-1 species studied in ref. 46. Due to their long excitation wavelength, the IPG class of molecular probes have been found to provide excellent signal-to-background ratio for single molecule microscopy, in studies undertaken in preparation for this work[46]. An illustrative diagram showing the system configured with the 488 nm laser is shown in Fig. 1, top left.

To obtain smooth excitation profiles over the image plane, the excitation laser beam is first spatially cleaned. For the 488 nm laser, it is first launched from the laser into a single mode optical fiber (Thorlabs P1-405B-FC-2) via a 20X microscope objective (Olympus PLN20X with 0.4 NA). At the other end of the fiber, a 10X objective (Olympus PLN10X) produces a parallel Gaussian beam of around 2 mm in diameter. The longitudinal positioning of this second objective on a micrometer stage allows for fine-tuning of the size of the illumination site on the sensing plane by controlling the beam divergence from the objective. For the 510 nm laser, the beam is instead launched into a 5x beam expander, with adjustable divergence, and then spatially filtered through a pinhole.

Immediately down-beam, a flip-mirror can be used to redirect the laser over a 4 m path to quantify its divergence. Optimal illumination performance was found to correspond to a very slightly divergent beam from this mirror. The excitation light is then passed through an adjustable iris to remove the halo. The 488 nm laser configuration then includes a 488 nm laser line filter (Thorlabs FL488-10); whereas for the 510 nm configuration, no subsequent excitation filtering proved to be necessary external to the vessel. The reflected ray from the line filter contains the long and short wavelength tails of the laser spectrum and is absorbed on an external beam dump.

The spatially and spectrally cleaned laser light is reflected from two external adjustable mirrors in a periscope arrangement to allow for fine steering of the path into the vacuum / pressure chamber. The beam next passes through a 1.6 mm thick sapphire pressure window into the vacuum or gas volume. Just in front of the sapphire window, a final 40 cm spherical planoconvex lens is placed at approximately its focal length from the main microscope objective, focusing the nearly parallel beam onto the back-focal plane.

Inside the chamber, the beam arrives at a fine-adjustment mirror with two degrees of freedom, which is set before the chamber is closed and left aligned for the duration of the experiment. This mirror directs the beam into an internal filter cube, which has emission and excitation filters and a long-pass dichroic beam splitter. For the 488 nm configuration, these are 500 nm short pass (SP) excitation (Thorlabs FESH050) and 500 nm long pass (LP) emission (Thorlabs FELH0500) filters, and a 490 nm dichroic mirror, also LP (Thorlabs DMLP490R). For 510 nm, a fluorescence microscopy filter set (Chroma 49023) consisting of an excitation filter: bandpass $500 \pm 20$ nm, emission filter: bandpass $560 \pm 25$ nm, and dichroic mirror: long pass 525 nm. On the fourth side of the filter cube, a high optical density neutral density filter serves as a shallow-depth internal beam dump.

The excitation beam is guided to the back face of a vented 100X, high NA microscope objective (Customized PLFLN100X; PLAN FLUOR 100X DRY OBJ, NA 0.95, WD 0.2), which we have customized for operation in high-pressure environments. Earlier experiments with commercial objectives resulted in various bursting and internal misalignment failures due to the pressurization and depressurization process. The beam is focused by the objective onto the front face of a 160 μm thick glass coverslip (Ted Pella Schott D263M 22 × 30 mm Glass Coverslips) in transmission mode, and the fluorescence emitted from the sample is collected back through the same high numerical aperture objective.

### Imaging

The longer wavelength fluorescence light emitted from the sample plane transmits through the dichroic mirror and exits through a second 2-inch diameter sapphire viewport (CeramTec 18617-01-CF). The infinity-space optical path on the fluorescence side is completely enclosed in black optical piping to prevent the ambient background from laboratory lighting. The light isolation in the image path is found to be sufficiently effective that single molecule resolution is comfortably accomplished with laboratory lights on. The image is reflected twice from a pair of mirrors in a periscope arrangement that allows for external adjustment of the region of the objective in the camera FOV. A second, external 500 nm LP filter (Thorlabs FELH050) removes any residual short-wavelength light, and then a tube lens focuses the image from infinity space onto the camera. Adjusting the tube lens position and focal length allows for the system to be run in modes with different levels of magnification. For this paper, we have used a medium level of magnification, with approximately 40 μm field of view per frame achieved using a 15 cm focal length tube lens.

The image acquisition device is an electron-multiplying CCD camera (Hamamatsu ImagEM2 EMCCD) with 90% quantum efficiency and a 512 × 512 array of 16 μm × 16 μm pixels (pix), operating at 500 ms exposure time. The camera is connected via a demountable tube coupling with an external shutter to protect the camera from being exposed to high-light environments. It is read out over IEEE 1394 conduits to a PC running a bespoke data acquisition software suite that we have developed specifically for this device that interfaces to the internal micrometer stages and camera readout.

## Alignment and focusing

At each $X$ and $Y$ position, there is an approximately 1 µm deep range of $Z$ values where the fluorescence plane is in focus. Finding this plane at each position requires the identification of a weak single molecule fluorescence signal among the range of plausible focal points, which vary by around 70 µm in absolute terms, given the repeatability afforded by our slide installation protocol. Few µm variations of the focal plane depth with temperature are observed, consistent with expectations based on the thermal expansion of structural materials and the range of temperatures recorded. Furthermore, when changing between gaseous and vacuum conditions we observe a slow drift of the focal plane position on few-hour timescales, consistent with the expected effect of swelling of plastics in the structure through gas absorption and degassing[63]. These effects, taken together imply that a dynamical method of establishing and tracking the focal plane is required.

To find the focal plane, a single-molecule-sensitive autofocus mechanism has been developed. The focal metric is the ratio of the pixel-wise kurtosis over the mean pixel intensity, which is maximized at each $X$ and $Y$ to establish the focal plane location in stage travel coordinates. This metric favors images with a small number of bright pixels among a field of dimmer ones, which is the characteristic of an in-focus image. Occasionally a cosmic ray muon or radioactive event passes through the EMCCD leading to one or a few extremely bright pixels, presenting to the autofocus metric as an anomalously in-focus image. We exclude such anomalous events by rejecting images with up to 5 bright pixels among a field of otherwise dim ones. This signature can also be distinguished from that associated with an extremely bright fluorescence emitter by its lack of persistence when stationary in the fluorescence plane.

Figure 7 illustrates the performance of the autofocus metric using single BODIPY molecules. Far from the focal plane, there are no discernible features, either in the images or in the metric. A set of very dim features are visible on the back of the slide. Since there is no fluorescent layer coated onto this back surface, we interpret these features as being associated with a small amount of out-of-focus fluorescence light scattering from optical imperfections on the reverse of the glass. Inside the cover-slip volume, no visible features are present. On the far side of the coverslip where we approach the desired focal plane, a sharp spike in autofocus metric is apparent. A higher resolution scan in this region (inset in Fig. 7, left) shows that the depth of focus is around 1 µm. Maximizing the autofocus metric is reliably found to obtain focal plane to ±0.3 µm precision.

After focusing, the images of single point-like emitters exhibit consistent point-spread function (PSF) in air, vacuum, and pressurized gases. Figure 8 shows the $X$ and $Y$ projections of the measured point-spread function obtained using single molecules within 7 bars of argon gas. These PSF projections are obtained by averaging over the $X$ and $Y$-directions around the brightest 20 fluorescent emitters, re-focusing in 5 distinct locations (Fig. 8, left). The width of the PSF depends slightly upon how perfectly the focal plane has been obtained, but in all cases is very similar to the expectation from Abbe's theory of diffracting optics[64]. The averaged PSFs in the $X$ and $Y$-directions are compared directly to the Abbe limit in Fig. 8, right. These results suggest that the microscope essentially saturates the theoretical limit of optical resolution, even when running within a pressurized noble gas environment. We note that, due to its single molecule sensitivity, super-resolution techniques can also be used in this system to advance beyond the diffraction limit, though this is of limited value for the barium tagging application.

Once a slide is installed, the standard procedure is to find the focal $Z$ position at 5–10 points on the face of the slide. These data are then used to generate a 3D map of the focal plane by extrapolating a 2D surface through the data points. This fitted focal plane can then be used to find the focus at any other point for subsequent imaging. The focal plane map continues to accumulate data points as in-focus images are found, which serves to continuously refine its precision for extrapolation to new points. This approach allows the focus to be found rapidly at new imaging locations and thus allows for the construction of larger, raster images by scanning over X-Y positions.

## Image sequence manipulation and analysis

Here we present a more complete description of the spot identification and analysis method. A figure showing some of the steps in the analysis chain is provided in Fig. 9. Raw data is represented by a sequence of $N_I$ images $I_i(x, y)$, indexed by integer $0 < i < N_I$, each composed of pixel values at $x, y$. The first processing step (Fig. 9a) is to remove diffuse background light and camera noise by applying a filter via convolution with two Gaussian functions of width $\sigma_A$ and $\sigma_B$ to obtain filtered image $J(x, y)$,

$$J(x, y) = I(x, y) \otimes \left[ \frac{1}{2\pi\sigma_A^2} \exp\left( -\frac{x^2 + y^2}{2\sigma_A^2} \right) - \frac{1}{2\pi\sigma_B^2} \exp\left( -\frac{x^2 + y^2}{2\sigma_B^2} \right) \right].$$

(1)

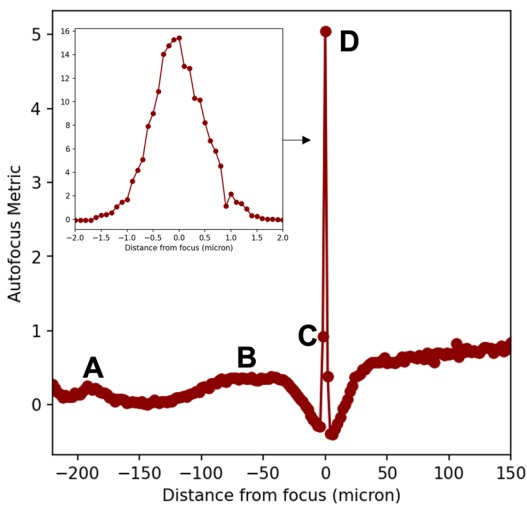

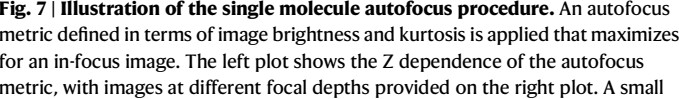

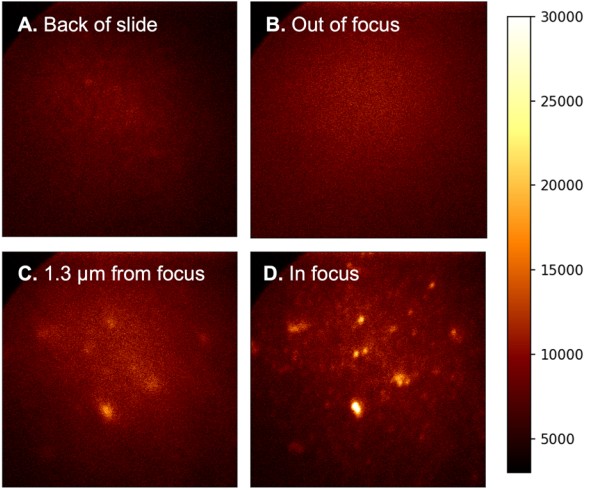

**Fig. 7 | Illustration of the single molecule autofocus procedure.** An autofocus metric defined in terms of image brightness and kurtosis is applied that maximizes for an in-focus image. The left plot shows the Z dependence of the autofocus metric, with images at different focal depths provided on the right plot. A small signal is seen on the back face of the slide (A). No features are present when totally out of focus (B). Within around 2 microns of focus, some activity is seen (C), with sharp images only at the focal plane (D). The inset shows that the depth of the in-focus region is around 1 µm.

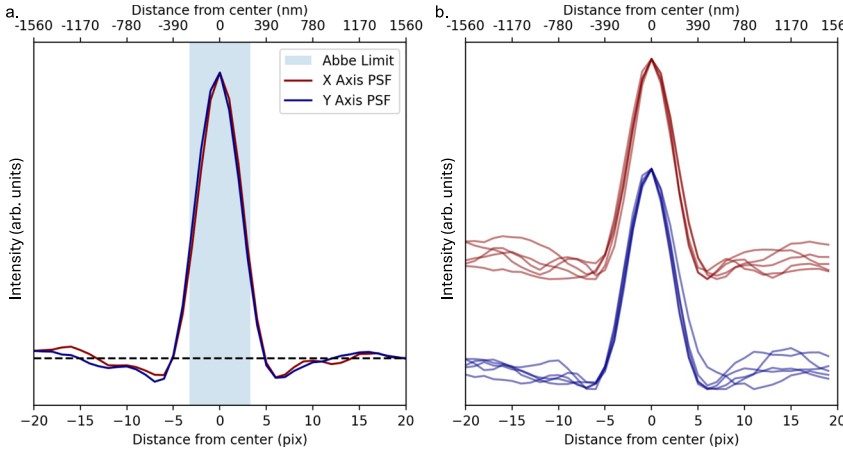

**Fig. 8 | Study of position resolution of the diffraction-limited microscope system. a** Measured point-spread function in *X* and *Y*-directions at five re-focused locations using single molecules in 7 bar argon gas. **b** Average X and Y PSF compared to the Abbe diffraction limit. The horizontal line on the right plot represents the baseline, as measured away from the bright spot.

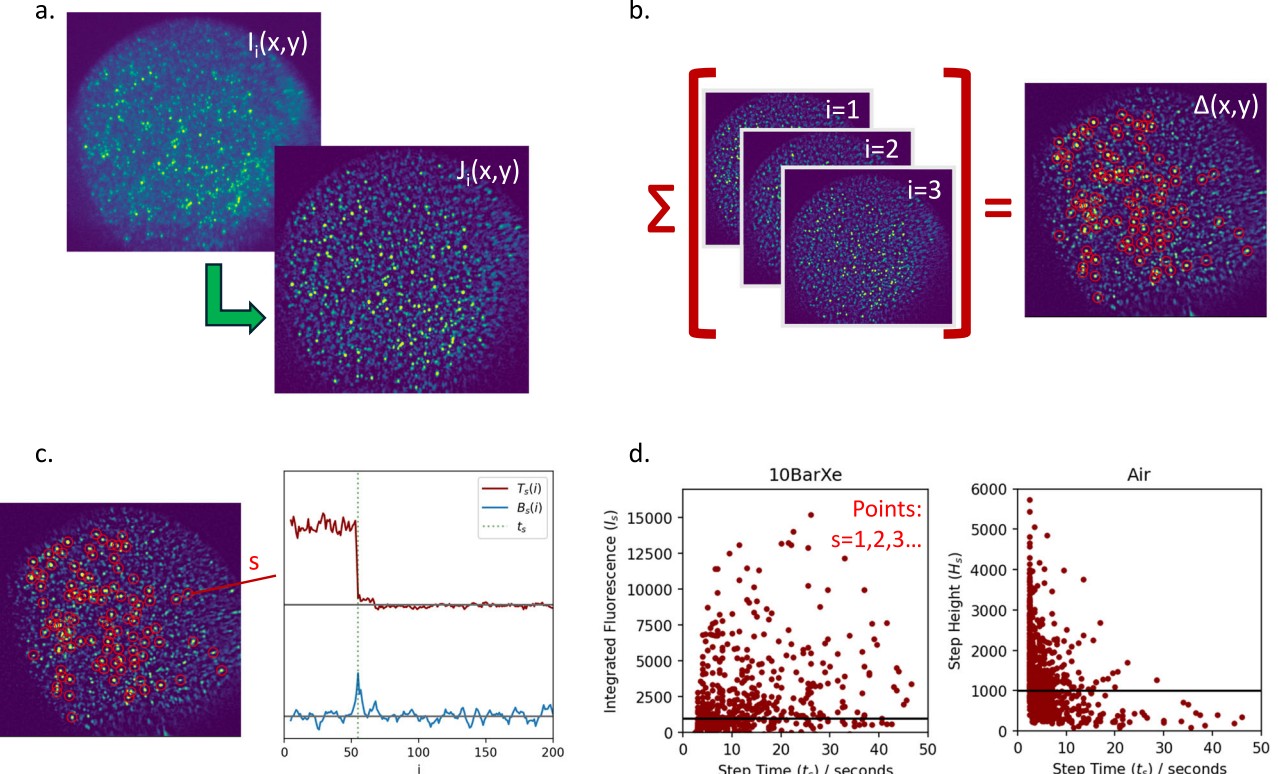

**Fig. 9 | Illustration of the steps in the image series analysis used to identify and quantify the behavior of single molecule candidates. a** Frequency filtering of raw data to remove diffuse backgrounds and CCD speckle noise. **b** Bright spot identification by pixels over the threshold. **c** Step time series identification. **d** Comparison between images in different conditions.

The equal normalization but opposite signs of the two Gaussian functions means that this filter is approximately peak-area conserving as long as the spot is smaller than $\sigma_A$ and $\sigma_B$, and it effectively acts as a local averaging of region $\sigma_A$ and background subtraction of region $\sigma_B$, removing the baseline light around each fluorescent spot; or frequency space it can be considered as serving as both a low- and high-pass filter. Based on trial and error, good performance is found with $\sigma_A \sim 4$ pixels and $\sigma_A \sim 8$ pixels, though the analysis results are rather insensitive to modest changes in these values. To identify points of interest, the first 20 and last 20 frames of each image are subtracted to produce a

difference map $\Delta_i(x, y)$ between the beginning and end of the sequence (Fig. 9b),

$$\Delta_i(x,y) = \frac{1}{20}\left[\sum_{i=1}^{20} J(x,y) - \sum_{i=N_I-20}^{N} J(x,y)\right] \quad (2)$$

The highest-valued pixels in $\Delta$ are identified as points of further interest. If a pixel is adjacent to either an identified point of interest or otherwise a previously vetoed pixel, it is itself vetoed, so that only new

isolated spots are added to the list $S$ of $N_S$ points of interest, indexed by integer $s$.

$$S = \{[x_s, y_s]\}, \quad 1 < s < N_S. \tag{3}$$

The first 100 points S are represented as red circles in (Fig. 9b). At each $S$, a time series $B$ is produced of the spot brightness over each of the $N_I$ frames (Fig. 9c).

$$B_s(i) = J_i(x_s, y_s), \qquad 1 < i < N_I. \tag{4}$$

This series is then analyzed to search for large steps that are consistent with single-molecule photobleaching events. A variable called the Step Confidence $C_s(i)$ is used to identify sharp steps, where

$$C_s(i) = \frac{\mu_-(i,s) - \mu_+(i,s)}{\sqrt{\sigma_-(i,s)\sigma_+(i,s)}}. \tag{5}$$

Here $\mu_\pm$ and $\sigma_\pm$ are the mean and standard deviation of the activity in the 5 frames before and after sample $i$, respectively,

$$\mu_-(i,s) = \frac{1}{5}\sum_{j=i-5}^{i} B_s(j), \quad \mu_+(i,s) = \frac{1}{5}\sum_{j=i}^{i+5} B_s(j), \tag{6}$$

$$\sigma_-(i,s) = \sqrt{\frac{1}{5}\sum_{j=i-5}^{i}\left(B_s(j)-\mu_-\right)^2}, \quad \sigma_+(i,s) = \sqrt{\frac{1}{5}\sum_{j=i}^{i+5}\left(B_s(j)-\mu_+\right)^2}. \tag{7}$$

A spike in the value of $C_s(i)$ represents a step that is larger than the typical fluctuations in the vicinity of that time. We accept steps that have $C_s(i) > C_{max} = 5$, though qualitatively similar results are found for any value in the range $3 < C_{max} < 10$. For cut values higher than 10, too few spots pass to furnish reasonable statistics for further analysis; cuts below 3 begin to accept noise from fluctuations that do not appear to be associated with single molecule photo-bleaching transitions in the camera images.

Once the steps of interest are identified, we consider their properties for further analysis. We define the step time $t_s$ for a given single ion candidate to be the time of the first $i$ in each image sequence where $C_s(i) > C_{max}$. The brightness of the spot that bleached is quantified via the Integrated Fluorescence $I_s$, defined via

$$I_s = \frac{1}{t_s}\sum_{i=1}^{t_s} B_s(i). \tag{8}$$

The distributions of $t_s$ and $I_s$ are shown in Fig. 9d. To report $t_s$ in physical units, it is multiplied by the camera exposure time of 500 ms.

Since some spots photo-blink in addition to photo-bleaching they may experience multiple transitions in one image sequence; as such these variables are not a unique or complete set of criterion by which to measure fluorescence activity. More detailed quantification of images sequences $I_i(x, y)$ and brightness traces $B_s(x, y)$ is surely possible. Nevertheless, we have found these methods to be functionally useful for identifying single ion candidates and for quantitatively comparing fluorescence responses in different conditions, the primary goals of the present work.

## Mechanical design and gas handling

The high-pressure microscope system is built into a 16-inch long, 6-inch diameter custom-manufactured pressure vessel with 8-inch ConFlat flanges on both ends. The microscope front end-cap is fixed down to an air-levitated optical table with a thick aluminum bracket. Vibration isolation was found to be crucial to achieving optimal resolution, and careful positioning of the various vacuum pumps and circulation pumps around the optical table proved to be instrumental in achieving sharp images. The pressure vessel pipe and back end cap slide on a set of two parallel rails to open and close the vessel, leaving the microscope head fixed in place in order to maintain rough alignment when opening and closing the large CF flanges.

Inside the vessel and cantilevered from the front end cap, a machined aluminum frame supports a small HDPE bracket that holds the internal microscope objective in a fixed position. A 3-axis vacuum stage system (PI Q545.140) is used to maneuver and monitor a microscope slide in three dimensions in front of the objective. The stage is rated for ultra-high vacuum and has nanometer precision along all three axes, with an active feedback loop inside the device. Our specifications demand only a few hundred-nanometer precision, below which diffraction limits the point spread function, and the stage comfortably meets these requirements. Careful frequency tuning of the feedback mechanism within the stages had to be made in order to avoid exciting resonant mechanical normal modes of the cantilevered system that inhibited stable positioning. The full range of motion of the stages is $\pm 6.5$ mm in each direction, though our control software restricts the $X$ and $Y$ (in-plane) motion to a region of $10 \times 10$ mm and the range of $Z$ (focusing) motion to within 1.5 mm in front of the microscope slide, to avoid damage from scratching the objective.

The system is evacuated using a turbo pumping station (Pfeiffer HiCube 80 Eco) which can be decoupled using an isolation valve (Carten HB-51) when pressurized. Despite the use of some plastics in the system, the vacuum quality, as monitored by a hot filament ion gauge (Kurt Lesker KJLC 354) routinely reaches $10^{-6}$ Torr prior to filling with gas, which is an adequate vacuum quality for subsequent fill and operation of time projection chamber detectors. Pressurized xenon gas is supplied to the system by a specially constructed gas handling system. The majority of the gas handling system is formed from 1/4 inch stainless steel piping with Swagelok fittings mounted to a gas control panel. The same system was used to supply purified xenon to the NEXT-CRAB0 detector in ref. 61.

Noble gases are supplied from cylinders with 99.999% purity and circulated through hot (SAES PS4-MT3-R-1) and cold (SAES MicroTorr HP190-902F) zeolite getters to clean at ~3 slpm for several hours to remove oxygen, water and nitrogen contamination. Pumping action is provided by a hermetic, piston-driven gas pump with neoprene buffers (PumpWorks PW2070) with pump speed controlled by a variac on the power line. Experience with devices on the same gas system shows that this is sufficient to achieve part-per-billion levels of oxygen and water impurity. Gas pressure is monitored by several analog pressure gauges on the gas panel, and over-pressurization is prevented by a 400 psig relief on the panel, 250 psig relief on the vessel, and a 15 psig burst disk on the vacuum line. The temperature inside the vessel is monitored by an internal thermocouple, and flow both into and out of the vessel in standard liters per minute is monitored using mass flow meters (Omega FM1800).

After running with argon, the gas is typically vented to the room through a vent line, whereas due to its much higher cost xenon is recaptured into bottles by cryogenic condensation with liquid nitrogen. A few psi of xenon is typically lost with each fill cycle due to the incompleteness of this capture process. A manifold-like gas mixing arrangement allows for the use of multiple gases or mixtures if needed.

## Data availability

High level processed data to reproduce plots from this manuscript can be found at the following https://doi.org/10.5281/zenodo.14043360. Original raw data can be obtained from the authors upon request.

## Code availability

Code to reproduce the figures in this manuscript is available at the following https://doi.org/10.5281/zenodo.14043360.

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

## Acknowledgements

This program of single barium ion sensing is part of a collaborative project to employ SMFI chemosensors functional at the solid-gas interface, as R&D toward barium tagging for the NEXT experiment. We acknowledge support from the US Department of Energy under awards DE-SC0019054 and DE-SC0019223, the US National Science Foundation under award number NSF CHE 2004111, and the Robert A Welch Foundation under award number Y-2031-20200401 (University of Texas Arlington). The NEXT Collaboration acknowledges support from the following agencies and institutions: the European Research Council (ERC) under Grant Agreement No. 951281-BOLD; the European Union's Framework Program for Research and Innovation Horizon 2020 (2014–2020) under Grant Agreement No. 957202-HIDDEN; the MCIN/AEI of Spain and ERDF A way of making Europe under grants RTI2018-095979 and PID2021-125475NB, the Severo Ochoa Program grant CEX2018-000867-S and the Ramón y Cajal program grant RYC-2015-18820; the Generalitat Valenciana of Spain under grants PROMETEO/2021/087 and CIDEGENT/2019/049; the Department of Education of the Basque Government of Spain under the predoctoral training program non-doctoral research personnel; the Portuguese FCT under project UID/FIS/04559/2020 to fund the activities of LIBPhys-UC; the Pazy Foundation (Israel) under grants 877040 and 877041; the US Department of Energy under contracts number DE-AC02-06CH11357 (Argonne National Laboratory), DE-AC02-07CH11359 (Fermi National Accelerator Laboratory). Finally, we are grateful to the Laboratorio Subterráneo de Canfranc for hosting and supporting the NEXT experiment.

## Author contributions

B.J.P.J. and D.R.N. conceived the project. N.K.B. advanced prototypes of the system that led to the successful realization of the present design. A.M. and B.J.P.J. designed and built the apparatus and wrote control software for the device. L.R.N., K.E.N., and E.D. contributed to the construction and alignment of the system and obtained preliminary data. B.J.P.J. collected the data and performed the analysis reported in this manuscript. F.W.F., R.M., and R.L.M. characterized the fluorescent dyes synthesized by the BODIPY molecules used for microscope calibration. The NEXT collaboration assisted with the editing and revision of the manuscript.

## Competing interests

The authors declare no competing interests.

## Additional information

# NEXT Collaboration

C. Adams[3], H. Almazán[4], V. Álvarez[5], B. Aparicio[6], A. I. Aranburu[7], L. Arazi[8], I. J. Arnquist[9], F. Auria-Luna[6], S. Ayet[10], C. D. R. Azevedo[11], J. E. Barcelon[7,12], K. Bailey[3], F. Ballester[5], M. del Barrio-Torregrosa[7], A. Bayo[13], J. M. Benlloch-Rodríguez[7], F. I. G. M. Borges[14], A. Brodolin[7,12], S. Cárcel[10], A. Castillo[7], S. Cebrián[15], E. Church[9], L. Cid[13], C. A. N. Conde[14], T. Contreras[16], F. P. Cossío[7,17], G. Díaz[18], T. Dickel[19], C. Echevarria[7], M. Elorza[7], J. Escada[14], R. Esteve[5], R. Felkai[8], L. M. P. Fernandes[20], P. Ferrario[17,21], A. L. Ferreira[11], Z. Freixa[17,21], J. García-Barrena[5], J. J. Gómez-Cadenas[17,21], R. González[7], J. W. R. Grocott[4], R. Guenette[4], J. Hauptman[22], C. A. O. Henriques[20], J. A. Hernando Morata[18], P. Herrero-Gómez[23], V. Herrero[5], C. Hervés Carrete[18], P. Ho[2], Y. Ifergan[8], F. Kellerer[10], L. Larizgoitia[7], A. Larumbe[6], P. Lebrun[24], F. Lopez[7], N. López-March[10], R. D. P. Mano[10], A. P. Marques[14], J. Martín-Albo[10], G. Martínez-Lema[8], M. Martínez-Vara[7], K. Mistry[1], J. Molina-Canteras[6], F. Monrabal[17,21], C. M. B. Monteiro[20], F. J. Mora[5], P. Novella[10], A. Nuñez[13], E. Oblak[7], J. Palacio[13], B. Palmeiro[4], A. Para[24], I. Parmaksiz[1], A. Pazos[17], J. Pelegrin[7], M. Pérez Maneiro[18], M. Querol[10], A. B. Redwine[8], J. Renner[18], I. Rivilla[7,21], C. Rogero[17], L. Rogers[3], B. Romeo[7], C. Romo-Luque[10], F. P. Santos[14], J. M. F. dos Santos[20], M. Seemann[7], I. Shomroni[23], P. A. O. C. Silva[20], A. Simón[7], S. R. Soleti[7], M. Sorel[10], J. Soto-Oton[10], J. M. R. Teixeira[20], S. Teruel-Pardo[10], J. F. Toledo[5], C. Tonnelé[7], J. Torrent[7,25], A. Trettin[4], A. Usón[10], P. R. G. Valle[7,17], J. F. C. A. Veloso[11], J. Waiton[4] & A. Yubero-Navarro[7]

[3]Argonne National Laboratory, Argonne, IL, USA. [4]Department of Physics and Astronomy, Manchester University, Manchester, United Kingdom. [5]Instituto de Instrumentación para Imagen Molecular (I3M), Centro Mixto CSIC - Universitat Politècnica de València, Camino de Vera s/n, Valencia, Spain. [6]Department of Organic Chemistry I, University of the Basque Country (UPV/EHU), Centro de Innovación en Química Avanzada (ORFEO-CINQA), San Sebastián / Donostia, Spain. [7]Donostia International Physics Center, BERC Basque Excellence Research Centre, Manuel de Lardizabal 4, San Sebastián / Donostia, Spain. [8]Unit of Nuclear Engineering, Faculty of Engineering Sciences, Ben-Gurion University of the Negev, P.O.B. 653 Beer-Sheva, Israel. [9]Pacific Northwest National Laboratory (PNNL), Richland, WA, USA. [10]Instituto de Física Corpuscular (IFIC), CSIC & Universitat de València, Calle Catedrático José Beltrán, 2, Paterna, Spain. [11]Institute of Nanostructures, Nanomodelling and Nanofabrication (i3N), Universidade de Aveiro, Campus de Santiago, Aveiro, Portugal. [12]Centro de Física de Materiales (CFM), CSIC & Universidad del Pais Vasco (UPV/EHU), Manuel de Lardizabal 5, San Sebastián / Donostia, Spain. [13]Laboratorio Subterráneo de Canfranc, Paseo de los Ayerbe s/n, Canfranc Estación, Spain. [14]LIP, Department of Physics, University of Coimbra, Coimbra, Portugal. [15]Centro de Astropartículas y Física de Altas Energías (CAPA), Universidad de Zaragoza, Calle Pedro Cerbuna, 12, Zaragoza, Spain. [16]Department of Physics, Harvard University, Cambridge, MA, USA. [17]Department of Applied Chemistry, Universidad del Pais Vasco (UPV/EHU), Manuel de Lardizabal 3, San Sebastián / Donostia, Spain. [18]Instituto Gallego de Física de Altas Energías, Univ. de Santiago de Compostela, Campus sur, Rúa Xosé María Suárez Núñez, s/n, Santiago de Compostela, Spain. [19]II. Physikalisches Institut, Justus-Liebig-Universitat Giessen, Giessen, Germany. [20]LIBPhys, Physics Department, University of Coimbra, Rua Larga, Coimbra, Portugal. [21]Ikerbasque (Basque Foundation for Science), Bilbao, Spain. [22]Department of Physics and Astronomy, Iowa State University, Ames, IA, USA. [23]Racah Institute of Physics, The Hebrew University of Jerusalem, Jerusalem, Israel. [24]Fermi National Accelerator Laboratory, Batavia, IL, USA. [25]Escola Politècnica Superior, Universitat de Girona, Av. Montilivi, s/n, Girona, Spain.

