## [Transparent Peer Review file · Nature Communications]

Fluorescence Imaging of Individual Ions and Molecules in Pressurized Noble Gases for Barium Tagging in ^{136}Xe

Corresponding Author: Dr Benjamin Jones

Version 0:

Reviewer comments:

Reviewer #1

(Remarks to the Author)

The authors of the manuscript "Fluorescence Imaging of Individual Ions and Molecules in Pressurized Noble Gases for Barium Tagging in Xe-136" (NCOMMS-24-35678-T) report on instrumental developments leading to the first observation of the Ba²⁺ ion inside a high-pressure Xe gas environment. This is a significant advancement towards the development of techniques to tag Ba-136 in Xe gas detectors.

High-efficiency Ba tagging would be a breakthrough for experiments searching for a hypothetical nuclear decay of Xe-136 called neutrinoless double-beta decay. The signature for such a decay would be the release of two electrons in the Xe with a summed energy equal to the Q-value of the reaction and the generation of the daughter isotope Ba-136. The reconstruction of the position and energy released by the electrons is the primary signature for such a process, and Xe time-projection chambers are widely used for this. Tagging the daughter isotope would provide an additional method to discriminate signal from background-like events. This would enhance the sensitivity of such searches, paving the way for future endeavours aiming for background-free searches of Xe-136 neutrinoless double-beta decay.

The search for neutrinoless double-beta decay is an extremely challenging field. State-of-the-art detector designs aim at operating tonnes of double-beta-decay candidate isotope in ultra-low background environments. In these searches, the background is constituted by radioactive contaminants in the detector materials. Additionally, the standard-model-allowed two-neutrino double-beta decay reaction represents an irreducible background that can only be mitigated by improving the energy resolution.

The search for neutrinoless double-beta decay is an internationally recognised scientific priority. It would prove that neutrinos and antineutrinos are two sides of the same entity, laying the groundwork for the development of a new theory of fermion masses and of the matter-antimatter asymmetry in our universe.

The results presented in this manuscript appear solid, and the methodology used to extract them is robust. The authors performed rigorous tests on control samples that strongly corroborate their findings. The content of the manuscript and its Supplementary Information is sufficient to understand and assess the work done. Given the importance of this subject and the results obtained, I certainly think that these findings deserve to be published on Nature. I congratulate the authors for these achievements.

My only concern with this manuscript relates to the lack of some crucial information needed by a non-expert reader to place the results in the correct context. In particular, the requirements for using Ba tagging in a real experiment are only briefly mentioned. To be effective, Ba tagging needs to have very high efficiency. In addition, it must be matched with the capability of accurately measuring the summed energy and position of the electrons emitted in decay and then precisely transporting potential daughter ions to the imaging device. Otherwise, the irreducible background due to the standard-model two-neutrino double-beta decay will create a large number of false positives, further reducing the efficiencies. The challenges involved in scaling a Ba-tagging technology to the level needed for tonne- or multi-tonne-scale experiments could also be discussed more concretely.

Thus, I would recommend the authors revise the introductory section to put their result in context, highlight the importance of the Ba-tagging detection integrated efficiency, and elaborate more later on the specific imaging efficiency obtained in this

specific work. While reviewing the introduction, I would also suggest providing more information about what neutrinoless double-beta decay is and what its experimental signature is. Finally, I had some difficulties following section 1 as I come from a particle physics background. I would suggest going through the text and adding a minimal introduction to the key techniques and technicalities discussed, as well as spelling out acronyms.

I have attached a list of detailed suggestions and comments below.

****Introduction****

1. I disagree with the statement that “all existing techniques to search for $0\nu\beta\beta$ have been limited by backgrounds from radiogenic activity in detector materials.” Some experiments have reached background levels below 1 background event in the ROI over their design exposure, while others are actually limited by the $2\nu\beta\beta$ rate in the ROI. Also, together with $2\nu\beta\beta$ events, solar neutrinos might be dominating the background budget in future liquid scintillator detectors.
2. The units used to quote the background index (“ $b < 0.1$ (ct ton ROI) $^{-1}$ ”) are unusual. I think it should be “ct/(ton keV yr).” Also, ROI has not been defined, nor is the concept that the primary signature of the decay is a monoenergetic energy release equal to the Q-value of the decay.
3. The introduction is missing a discussion about what neutrinoless double-beta decay is (two neutrinos in two protons + electrons) and what its experimental signature is, particularly the peak at the Q-value.
4. Non-expert readers will not be able to understand the discussion about normal and inverted neutrino mass ordering. This should be expanded. The effective Majorana mass is not mentioned. Furthermore, the discussion about the normal ordering being below the inverted ordering parameter space is misleading. The normal ordering parameter space expands all the way throughout the inverted ordering one and is already being explored by the running experiments.
5. Please expand the discussion about what Ba tagging requires, including ion trapping and transportation. Requirements in terms of efficiencies and resolutions could be detailed. Additionally, any new hardware introduced in the setup will bring additional new background. What are the radiopurity requirements for the ion imaging hardware discussed in this manuscript?

****Section 1****

6. The terms step time and photo-bleaching time are not defined.
7. SI, IPG, and other acronyms are not defined.
8. From the brief discussion in the text, I could not really understand the origin of the streaks in Fig 2. Could you please elaborate on this? Also, please comment on what is the single-ion detection efficiency? Are the streaks reducing it?

****Conclusions****

I would recommend elaborating on what the radioactive requirements for operating the hardware in the final setup are and mentioning explicitly the envisioned size of the area that will need to be monitored. This will give an idea of the scaling required from the mm^2 achieved in this work.

Reviewer #2

(Remarks to the Author)

Reviewer #3

(Remarks to the Author)

This study aims to demonstrate the feasibility of the detection of single barium ions in a high-pressure xenon gas environment, towards the advancement in the search for neutrinoless double beta decay ($0\nu\beta\beta$). The detection of the $0\nu\beta\beta$ provides insights into the Majorana nature of neutrinos, with significant implications for our understanding of the universe's matter-antimatter asymmetry and the origin of neutrino mass. Identifying Ba^{2+} ions resulting from the double beta decay of Xe-136 is technologically challenging, but could be an important background mitigation technique for future (multi-) ton-scale $0\nu\beta\beta$ experiments. The work involved the development of a high-pressure fluorescence imaging system utilizing molecular chemosensors that become fluorescent upon chelation with Ba^{2+} ions. The study successfully demonstrated the ability to resolve individual Ba^{2+} ions at the gas-solid interface, marking a step toward achieving background-free sensitivity in $0\nu\beta\beta$ experiments with Xe-136.

The experimental setup featured a diffraction-limited microscope mounted inside a high-pressure chamber with a xenon gas recirculation system to maintain high xenon purity. Key findings demonstrated the capability to identify single ions in a high-pressure environment and revealed distinct photobleaching behaviors of the fluorophores in xenon compared to air,

highlighting the significant role of oxygen in this process. This paper marks an important step forward in the ongoing effort of the NEXT collaboration to develop Ba-tagging techniques for future large-scale $0\nu\beta\beta$ experiments using pressurized gaseous xenon time projection chambers.

Major comments

The novelty of this work, particularly in comparison to previous efforts and results by the NEXT collaboration, could be more explicitly stated. A clear distinction in technological and methodological advancements should be more prominently highlighted to underscore the potential breakthrough nature of this research.

This manuscript describes a rather technical study that is one step in a series of steps required to identify the Ba^{2+} ion in the hypothetical neutrinoless double beta decay and cannot be seen as a major scientific advancement as such. The manuscript should be published, but a more technical Journal than Nature Communications would be more appropriate.

Minor comments

-Somewhere in the introduction it would be useful to explicitly mention the $0\nu\beta\beta$ reaction producing Ba^{2+} , e.g. $Xe-136 \rightarrow Ba-136^{2+} + 2e^-$

-The background index is mentioned as $b < 0.1(\text{ct ton ROI})^{-1}$, but this has wrong units (ct should be in the numerator), this should also be expressed in terms of exposure, i.e., ton-yr and ROI should be defined.

-FIG 1 and caption:

Minor spellings: "pressure chamber.r" -> "pressure chamber." The label "XYZ piezeo stages" -> "XYZ piezo stages." To improve readability, consider horizontal text or a vertical orientation of the entire figure. The acronyms "LP" and "SP" should be fully written out in the figure or the caption to ensure that all readers clearly understand their meaning (it is only defined in the SI).

-The different patterns observed in the processed image shown in FIG. 3 are properly introduced, however the distinct features present in the left panel of FIG. 2—specifically, the straight lines with more dense bright points in two different parts of the image and the very intense blobs—are not addressed or discussed in either the text or the figure caption. Including an explanation or discussion of these features would enhance the reader's understanding and interpretation of the data presented.

-The horizontal line in the right panel of FIG. 4 is not explained in the text or the figure caption. It is unclear what this line represents.

-The caption of FIG. 4 mentions that a single slide was used in the right figure, but this raises questions about the representativeness of the data. The manuscript should better describe how this slide was chosen. Was it selected based on certain criteria or randomly?

-This point leads to a broader question: what criteria are used to determine that a bright spot is a signature of Ba^{2+} ? The text mentions that "a small number of very weak emitters are present in the Ba^{2+} -free samples, though the bright spots associated with Ba^{2+} -bound IPG molecules are unambiguously identified as being present only in the Ba^{2+} -chelated sample." However, it is not clear how the distinction is made between Ba^{2+} -bound and Ba^{2+} -free samples. Is the differentiation based only on the intensity of the peak?

The statement, "A cut on the maximal identified step height in each trace is used to identify single molecule photo-bleaching event candidates," partly answers the question but it would be helpful to provide more details on this cut or even include a plot, to better illustrate the criteria used for identifying single molecule photo-bleaching events. This additional information would enhance the clarity and robustness of the presented data.

-Right panel of Figure 6, the second plot (from the top) does not appear to represent a single barium ion chelated with IPG-1 as claimed. The presence of secondary peaks with large brightness suggests that multiple barium ions may be chelated rather than a single ion. Some clarification on this particular event would be beneficial to understand the observed data better and to confirm whether the fluorescence signal corresponds to a single barium ion or multiple ions.

-In the sentence, "Such candidates are identifiable in all conditions tested, including ambient air, vacuum and and pressurized argon," remove 2nd "and"

-At the end of the first paragraph of Methods III.A has a period missing.

Reviewer #4

(Remarks to the Author)

Meaningful measurements are being made.

It would be worthy of publication if the minor corrections and responses below were made.

p2

- The unit should be corrected
 $b < 0.1 \text{ (ct ton ROI)}^{-1} \rightarrow b < 0.1 \text{ ct (yr ton ROI)}^{-1}$

p3

- wrong article number
[23] Journal of High Energy Physics 2018, 1 (2018) \rightarrow Journal of High Energy Physics 2018, 112 (2018)
- What is the Ba^{2+} remaining efficiency ?
Even if recombination is low, it cannot be said that 100% of Ba^{2+} will survive.
There is a possibility that the charge is distributed by collisions with xenon atoms or impurities.
Are there any measured values or literature values for the Ba^{2+} survival efficiency?

p4

- Typo in FIG.1 caption
chamber.r \rightarrow chamber.
- Explain abbreviation if you use in the latter text
Supplementary Information \rightarrow Supplementary Information (SI)

p5

- unit is not italic for "33 μm "
- In Fig.2, specify why lattice pattern inefficient region exist and it is affect to the Ba^{2+} tag measurement or not.
- Match the expression in the text in Fig2. caption
single molecule fluorescence \rightarrow single molecule fluorescence candidate
- Put the condition of the measurement of Fig.2.
Do you put Ba^{2+} to the coverslip uniformly?
- What is the time for 1 step scan for 1 mm^2 area?

p6

- In Fig.3 (right), some bright clusters seem to cut out at the edge of the 33 μm boxel. Why does it comes from?
- In Fig.4 (right), specify black horizontal line.
- In Fig.4 (right), why there is no event in the range of 0~3 and 46~50 of Step Time?
- What is the reason of existence of weak emitters in the IPG only sample?

p7

- In the text, Fig.6 is mentioned before Fig.5.
If there is no reason, it is better to swap them.
- In Fig.5, why 1st step bin count is different in 10barXe and Air?
Is it just due to the amount of coating?
- What is the considered reason of the photo-blinking?

p8

- Mean life times are inconsistent to that are written in Fig.5
- What is the 15GB of data size correspond to? 1shot of the 1 mm^2 image?

Reader cannot judge 15GB is large or not without any reference.

p14

- typo: have have  have

Version 1:

Reviewer comments:

Reviewer #1

(Remarks to the Author)

The authors have addressed all my concerns. I am particularly pleased with the new introduction, which clearly addresses aspects of Ba tagging that I have long wondered about while reading previous publications from the NEXT collaboration. Below, I list a few very minor comments on the introduction for the authors' consideration. However, I am already able to recommend the acceptance of this article in Nature and do not require reviewing the revised version.

Minor comments:

- Reference 7 is more than 20 years old and outdated. There are far more recent reviews available.

- The paper by Schechter and Valle has some limitations. In particular, the diagram proposed in that work would generate only a very small Majorana mass (on the order of $1e-28$ eV), which is extremely so tiny compared to the experimentally estimated values from oscillation experiments. See, for instance, JHEP 06 (2011) 091.

Double-beta decay specifically refers to neutrons converting to protons and leptons, not protons converting to neutrons. Should the authors want to mention it, double-electron capture is the other process that is actively experimentally investigated.

- The sentence "likely the only observed manifestation of physics above the electroweak scale" is not trivial, and I am unsure whether it is correct. Please provide a reference to support this claim.

- Please include the confidence level of the half-life limit for Xe.

- The rate of $0\nu\beta\beta$ decay depends on several other factors, particularly the Majorana phases, which are currently the primary source of uncertainty along with the lightest neutrino mass eigenstate. Nuclear matrix elements and the phase space factor may also be worth mentioning.

- In the sentence "loose spatio-temporal requirement," would it be possible to quantify the spatial resolution? This is something I have always wondered about but have never seen addressed. Is there any specific work that quantifies this requirement that could be cited?

- The process of Ba capture and subsequent detection is complex, and the authors rightly state that the compounded efficiency requirements are high. They point out some work discussing capture efficiencies, but perhaps the addition of a small comment focusing on the imaging/detection efficiency, as this is the topic of the article, will help convince the reader that such efficiency requirements might be achievable with the proposed methods.

I would suggest proofreading the manuscript, as I noticed a few typos. For example, in the conclusions, "A rational zero suppression algorithm and a form of online trigger for frames of interest will be advantageous..."

Reviewer #2

(Remarks to the Author)

Reviewer #3

(Remarks to the Author)

We thank the authors for thoroughly addressing our comments and are satisfied with all answers. The manuscript has become more approachable to a wider audience and we think that the authors have provided convincing arguments for publishing this paper in Nature Communications. We therefore recommend acceptance.

Reviewer #4

(Remarks to the Author)

The authors have made most of the necessary corrections and clarifications.

> We suspect these are dim fluorescence associated with unbound IPG molecules interacting with the glass surface. They are below the cut which we apply to identify single molecule candidates so can be considered as a harmless background, for present purposes.

Some points in most right figure of Fig.4 seem to be over the cut threshold (1000).

If the threshold were higher (like 4000), that explanation would make sense.

There is very minor comment about the reference of [63] Thorlabs FEHH0500,

I can not find the product with this model number.

I wonder this is typo of FESH0500.

We sincerely thank the reviewers for their excellent feedback and overall very positive reviews. Several typographical errors were identified, which have now been fixed. The suggestions for further detail and clarification were all valid and valuable as well, and addressing them has significantly improved the readability of the manuscript.

We respond point-by-point below.

REVIEWER COMMENTS

Reviewer #1 (Remarks to the Author):

The authors of the manuscript “Fluorescence Imaging of Individual Ions and Molecules in Pressurized Noble Gases for Barium Tagging in Xe-136” (NCOMMS-24-35678-T) report on instrumental developments leading to the first observation of the Ba²⁺ ion inside a high-pressure Xe gas environment. This is a significant advancement towards the development of techniques to tag Ba-136 in Xe gas detectors.

High-efficiency Ba tagging would be a breakthrough for experiments searching for a hypothetical nuclear decay of Xe-136 called neutrinoless double-beta decay. The signature for such a decay would be the release of two electrons in the Xe with a summed energy equal to the Q-value of the reaction and the generation of the daughter isotope Ba-136. The reconstruction of the position and energy released by the electrons is the primary signature for such a process, and Xe time-projection chambers are widely used for this. Tagging the daughter isotope would provide an additional method to discriminate signal from background-like events. This would enhance the sensitivity of such searches, paving the way for future endeavours aiming for background-free searches of Xe-136 neutrinoless double-beta decay.

The search for neutrinoless double-beta decay is an extremely challenging field. State-of-the-art detector designs aim at operating tonnes of double-beta-decay candidate isotope in ultra-low background environments. In these searches, the background is constituted by radioactive contaminants in the detector materials. Additionally, the standard-model-allowed two-neutrino double-beta decay reaction represents an irreducible background that can only be mitigated by improving the energy resolution.

The search for neutrinoless double-beta decay is an internationally recognised scientific priority. It would prove that neutrinos and antineutrinos are two sides of the same entity,

laying the groundwork for the development of a new theory of fermion masses and of the matter-antimatter asymmetry in our universe.

The results presented in this manuscript appear solid, and the methodology used to extract them is robust. The authors performed rigorous tests on control samples that strongly corroborate their findings. The content of the manuscript and its Supplementary Information is sufficient to understand and assess the work done. Given the importance of this subject and the results obtained, I certainly think that these findings deserve to be published on Nature. I congratulate the authors for these achievements.

We thank the reviewer for this positive appraisal of our work and our manuscript.

My only concern with this manuscript relates to the lack of some crucial information needed by a non-expert reader to place the results in the correct context. In particular, the requirements for using Ba tagging in a real experiment are only briefly mentioned. To be effective, Ba tagging needs to have very high efficiency. In addition, it must be matched with the capability of accurately measuring the summed energy and position of the electrons emitted in decay and then precisely transporting potential daughter ions to the imaging device. Otherwise, the irreducible background due to the standard-model two-neutrino double-beta decay will create a large number of false positives, further reducing the efficiencies. The challenges involved in scaling a Ba-tagging technology to the level needed for tonne- or multi-tonne-scale experiments could also be discussed more concretely.

Thus, I would recommend the authors revise the introductory section to put their result in context, highlight the importance of the Ba-tagging detection integrated efficiency, and elaborate more later on the specific imaging efficiency obtained in this specific work. While reviewing the introduction, I would also suggest providing more information about what neutrinoless double-beta decay is and what its experimental signature is. Finally, I had some difficulties following section 1 as I come from a particle physics background. I would suggest going through the text and adding a minimal introduction to the key techniques and technicalities discussed, as well as spelling out acronyms.

We have significantly improved the introduction. We have given much extra context and extra material concerning barium tagging and neutrinoless double beta decay, as will be addressed in the more specific points below. In addition to this we have opted in the latest draft to present the results more clearly in the context of SMFI technology, as well as of barium tagging / $0\nu\beta\beta$, as embodied in the new introductory paragraph:

Single molecule fluorescence imaging (SMFI) is a Nobel Prize winning technique [ref] that has enabled major advances in biochemistry and cellular imaging [ref]. SMFI enables super-resolution microscopy [ref], illuminating features in cells far below the diffraction limit. In addition to transformationally advancing the resolution of optical microscopes, SMFI also represents the ultimate frontier in analytic chemistry. By custom chemosensor design, single molecules of specific analytes can be sensed both in-vitro and in-vivo [ref]. SMFI implemented at the gas solid interface has the potential to open a host of new applications, though beyond state-of-the-art microscopy methods, molecular and supra-molecular synthesis approaches are required for its realization. An especially compelling application that is currently driving the development of SMFI at the gas-solid interface is barium tagging [ref], the sensing of individual ions of Ba^{2+} in xenon, which could dramatically increase the discovery reach of neutrinoless double beta decay ($0\nu\beta\beta$) searches. This paper presents the first ever identification of individual ions at a high pressure gas interface, an important advance for both SMFI and for $0\nu\beta\beta$.

More information on the specific points follows.

I have attached a list of detailed suggestions and comments below.

****Introduction****

1. I disagree with the statement that “all existing techniques to search for $0\nu\beta\beta$ have been limited by backgrounds from radiogenic activity in detector materials.” Some experiments have reached background levels below 1 background event in the ROI over their design exposure, while others are actually limited by the $2\nu\beta\beta$ rate in the ROI. Also, together with $2\nu\beta\beta$ events, solar neutrinos might be dominating the background budget in future liquid scintillator detectors.

We agree with the referee, this was an improper generalization on our part and we now have clarified:

Two neutrino double beta decays ($2\nu\beta\beta$) are one source of background to $0\nu\beta\beta$, and can be efficiently rejected by technologies with full-width-half-maximum (FWHM) energy resolution $E_{FWHM} \leq 1\%$. The remaining backgrounds in contemporary experiments originate from radiogenic gamma rays in detector materials, cosmogenic material activation, and in principle, solar neutrino interactions in the detector. One especially

promising technical approach to remove all such events and hence reach the ultra-low background limit is “barium tagging”--- identification of the ^{136}Ba daughter ion produced in the double beta decay of ^{136}Xe ~\cite{Moe:1991ik}.

2. The units used to quote the background index (“ $b < 0.1(\text{ct ton ROI})^{-1}$ ”) are unusual. I think it should be “ $\text{ct}/(\text{ton keV yr})$.” Also, ROI has not been defined, nor is the concept that the primary signature of the decay is a monoenergetic energy release equal to the Q-value of the decay.

The unit had a typo (also pointed out by reviewer 3) that is now fixed, to $\text{ct}(\text{ton yr ROI})^{-1}$. Note we opt for the unit without keV in the denominator, following for example, this report:

https://science.osti.gov/-/media/np/nsac/pdf/docs/2016/NLDBD_Report_2015_Final_Nov18.pdf

Sometimes $\text{ct}/(\text{ton keV yr})$ is used, but since different techniques provide different energy resolutions, this does not allow for easy comparison across experiments using different methods to reject backgrounds (for example, germanium and bolometers primarily use energy resolution, whereas TPCs also add topological information to the event selection cuts). On the other hand, the quantity in units we have used is one that maps more directly to the ultimate sensitivity. We have cited the report above where we state this, to provide precedence for the unit:

detectors with ≥ 1 Ton of the double beta decay isotope and background levels (measured in counts per ton-year in the energy region of interest (ROI)~\cite{Irp0nubb}) of order $b < 0.1 \text{ ct}(\text{ton yr ROI})^{-1}$ are required~\cite{agostini2017discovery}.

3. The introduction is missing a discussion about what neutrinoless double-beta decay is (two neutrinos in two protons + electrons) and what its experimental signature is, particularly the peak at the Q-value.

This has been added:

One isotope that has been used in many $0\nu\beta\beta$ searches to date is ^{136}Xe , which can decay to ^{136}Ba via $^{136}\text{Xe} \rightarrow ^{136}\text{Ba} + 2e^{-}$. Because the daughter nucleus in the final state is very heavy relative to the electrons, they carry away almost all of the

available energy, producing a nearly mono-energetic line at the Q-value for the decay, $Q_{\beta\beta}=2457.8$ keV. Reconstructing electron energy deposits in media enriched in ^{136}Xe has enabled searches for this process, with the current strongest limit being 2.3×10^{26} yr~\cite{abe2023search}.

4. Non-expert readers will not be able to understand the discussion about normal and inverted neutrino mass ordering. This should be expanded. The effective Majorana mass is not mentioned. Furthermore, the discussion about the normal ordering being below the inverted ordering parameter space is misleading. The normal ordering parameter space expands all the way throughout the inverted ordering one and is already being explored by the running experiments.

We had originally opted to omit a lot of this discussion, in part to avoid overwhelming a general reader with neutrino physics, but we see the reviewers point that without it, some of the following the discussion of the mass ordering would be unclear. Below represents our approach to describing the salient points in a clear way for a general readership:

The expected rate of $0\nu\beta\beta$ depends on the neutrino masses and mixing parameters. Neutrino oscillations have measured two characteristic mass-squared differences between neutrino states, $\Delta m^2_{32}\sim 2.4\times 10^{-3}$ eV 2 and $\Delta m^2_{21}\sim 7.4\times 10^{-5}$ eV 2 ~\cite{navas2024review}. The three neutrino masses are thus either organized with a large gap between the heavier pair (normal ordering) or the lightest pair (inverted ordering). If the value of the lightest neutrino mass is smaller than 100 meV, the expected range of lifetimes for $0\nu\beta\beta$ depends strongly on this ordering. In the inverted case, or if the lightest neutrino mass is greater than 100 meV under the normal ordering, $0\nu\beta\beta$ ought to be discoverable with a lifetime $\tau \leq 10^{28}$ yr. To achieve such sensitivities...

5. Please expand the discussion about what Ba tagging requires, including ion trapping and transportation. Requirements in terms of efficiencies and resolutions could be detailed. Additionally, any new hardware introduced in the setup will bring additional new background. What are the radiopurity requirements for the ion imaging hardware discussed in this manuscript?

Material has been added in several places to address this comment in the latest version. First, we discuss barium tagging requirements in the introduction:

An efficient and selective barium ion tag could reduce contamination from all backgrounds except for $2\nu\beta\beta$ to effectively zero. Demonstration of a method of capture and imaging of barium ions from one to several tons of xenon requires significant advances in instrumentation. To provide either a significant sensitivity boost or a signal confirmation, barium ions must be captured and then identified with high efficiency (greater than around 50% to avoid substantial loss of sensitivity relative to current analysis methods~\cite{martin2016sensitivity}) in coincidence with electron energy deposits near the Q-value reconstructed with resolution better than 1% FWHM to reject the two-neutrino mode. The two-neutrino decay mode also produces barium ions at a rate of around 5 per kilogram per day~\cite{novella2022measurement}, which imposes a loose spatio-temporal requirement on the coincidences that must be established between the ion and electron signatures. 3D imaging of electron tracks may provide a further confirmation of the two-electron topology~\cite{ferrario2016first,simon2022boosting}, in principle enabling a robust three-fold coincident signature.

We have opted *not* to be more specific about the detailed implementation of the coincidence mechanism, as there is an allowed trade-off between spatial and temporal aspects, as well with with how efficiently $2\nu\beta\beta$ are reconstructed (each time a $2\nu\beta\beta$ is identified this can be used to “veto” a barium ion, for example), and different configurations under exploration in NEXT would meet these needs in different ways. However, the coincidence requirement is rather loose and hence appears comfortably achievable, so we hope the above is specific enough to outline the performance criteria.

In the conclusion we also comment on radiopurity and coupling to ion transportation methods:

Since barium tagging rejects all radiogenic background events, the radio-purity requirements of the molecular monolayer and barium imaging system are expected to be modest, and likely already to be met by the current system. Furthermore, if coupled with an ion transport device such as a radiofrequency carpet~\cite{jones2022dynamics}, the demonstrated scan area will also suffice for a realistic barium tagging sensor for a $0\nu\beta\beta$ experiment.

****Section 1****

6. The terms step time and photo-bleaching time are not defined.

Added:

“The \$\$\$ axis corresponds to the step time, which is defined as the time when the most significant change in fluorescence intensity between five pre-samples and five post-samples is observed.”

Added:

“ On the other hand, fluorescent molecules which are exposed to large integrated light intensities undergo destructive “photobleaching” reactions~\cite{demchenko2020photobleaching}, limiting the practically usable laser power for prolonged observation.”

7. SI, IPG, and other acronyms are not defined.

Expanded IPG = Ion Potassium Green and replaced SI with Methods section

8. From the brief discussion in the text, I could not really understand the origin of the streaks in Fig 2. Could you please elaborate on this? Also, please comment on what is the single-ion detection efficiency? Are the streaks reducing it?

There is some subtlety involved in defining the efficiency in different experimental conditions. Here we have applied Ba²⁺ in aqueous salt solution, whereas in 0nubb it will arrive in a xenon solvation shell around Ba²⁺ (see Ref 20, Bainglass et al). The efficiency of binding in the current configuration can be inferred based on reaction constants of the relevant chemical equilibria. The dissociation constant for IPG-1 is around 50mM (https://ionbiosciences.com/store/ipg1_tma/). This implies that for an equal mixture of Ba²⁺ and IPG, ~95% will be bound, ie it implies a 95% efficiency, for the studies in this paper. Our past work with mixtures (cited in Ref 28) has verified the expected binding constants in solution barium / IPG mixtures. In the current experiments we have over-saturated the solution in Ba²⁺ to maximize the observable signal, since our goal is to demonstrate single ion microscopy in pressurized gasses. On the other hand, the direct capture out of gas is not a dynamical equilibrium, and as long as an ion finds its way to a receptor (achievable with a densely packed monolayer) still higher efficiencies can be expected. Future work will measure the capture efficiency from the gas directly, but it has not been done here. We have a preference not to discuss these issues in this paper, since the comparison between different conditions is somewhat complex, and it could be seen as over-claiming to state that the high effective efficiencies from our solution tests is reflective of those gas-phase reactions. If

the reviewer feels very strongly, however, then some of the above discussion could be added - please let us know if this is still required, following this context.

The streaks are now better explained in the text, as follows:

Some visible streaks in the distribution of single molecule candidates emanate in a radial direction and are a result of the spin-coating protocol, where the solution from which molecules are deposited leaves some residual droplets as it dries.

We do not believe they have any impact on the imaging efficiency, as the emitting single molecules are still individually resolvable within them.

****Conclusions****

I would recommend elaborating on what the radioactive requirements for operating the hardware in the final setup are and mentioning explicitly the envisioned size of the area that will need to be monitored. This will give an idea of the scaling required from the mm² achieved in this work.

We have added:

“Since barium tagging rejects all radiogenic background events, the radio-purity requirements of the molecular monolayer and barium imaging system are expected to be modest, and likely already to be met by the current device. Furthermore, if coupled with an ion transport system such as a radiofrequency carpet~\cite{jones2022dynamics}, the demonstrated scan area will also suffice for a realistic barium tagging sensor for a $^{0\nu}\beta\beta$ experiment. Some modifications are still required for implementation with within a time projection chamber, which we now briefly discuss.... [and then continue with what we had in this section before]”

Reviewer #2 (Remarks to the Author):

Reviewer #3 (Remarks to the Author):

This study aims to demonstrate the feasibility of the detection of single barium ions in a high-pressure xenon gas environment, towards the advancement in the search for neutrinoless double beta decay ($0\nu\beta\beta$). The detection of the $0\nu\beta\beta$ provides insights into the Majorana nature of neutrinos, with significant implications for our understanding of the universe's matter-antimatter asymmetry and the origin of neutrino mass. Identifying Ba^{2+} ions resulting from the double beta decay of Xe-136 is technologically challenging, but could be an important background mitigation technique for future (multi-) ton-scale $0\nu\beta\beta$ experiments. The work involved the development of a high-pressure fluorescence imaging system utilizing molecular chemosensors that become fluorescent upon chelation with Ba^{2+} ions. The study successfully demonstrated the ability to resolve individual Ba^{2+} ions at the gas-solid interface, marking a step toward achieving background-free sensitivity in $0\nu\beta\beta$ experiments with Xe-136.

The experimental setup featured a diffraction-limited microscope mounted inside a high-pressure chamber with a xenon gas recirculation system to maintain high xenon purity. Key findings demonstrated the capability to identify single ions in a high-pressure environment and revealed distinct photobleaching behaviors of the fluorophores in xenon compared to air, highlighting the significant role of oxygen in this process. This paper marks an important step forward in the ongoing effort of the NEXT collaboration to develop Ba-tagging techniques for future large-scale $0\nu\beta\beta$ experiments using pressurized gaseous xenon time projection chambers.

Major comments

The novelty of this work, particularly in comparison to previous efforts and results by the NEXT collaboration, could be more explicitly stated. A clear distinction in technological and methodological advancements should be more prominently highlighted to underscore the potential breakthrough nature of this research.

Thank you for this comment - indeed, we should have been clearer about what makes this particular step both difficult and important. We summarize the past work by NEXT in the introduction as in v1 as before, but we now clarify better the nature of the advance in the text:

This paper presents the first ever demonstration of single Ba^{2+} imaging within the working medium of a time projection chamber.

And

An especially compelling application that is currently driving the development of SMFI at the gas-solid interface is barium tagging

~\cite{Moe:1991ik,nygren2016detection}, the sensing of individual ions of Ba^{2+} in xenon, which could dramatically increase the discovery reach of neutrinoless double beta decay ($0\nu\beta\beta$) searches. This paper presents the first ever identification of individual ions at a high pressure gas interface, an important advance for both SMFI and for $0\nu\beta\beta$.

Please see also the added context about SMFI provided in response to the Reviewer 1, which is also relevant to this point.

This manuscript describes a rather technical study that is one step in a series of steps required to identify the Ba^{2+} ion in the hypothetical neutrinoless double beta decay and cannot be seen as a major scientific advancement as such. The manuscript should be published, but a more technical Journal than Nature Communications would be more appropriate.

We do understand the reviewers instincts here, since advances concerning realization of novel microscope imaging modalities are often not represented in Nature Journals. However, in this case the work represents a major step toward an important goal for the field of neutrinoless double beta decay, and a substantial advance in single molecule fluorescence imaging more generally. Other steps of this magnitude have been published in Nature journals. For example, consider Ref. <https://www.nature.com/articles/s41586-019-1169-4>. That paper was not the first single barium imaging demonstration (the work in <https://journals.aps.org/prl/abstract/10.1103/PhysRevLett.120.132504> pre-dated it), but rather, it realized single atom imaging in an innovative new way - we agree that their advance was substantial, and met the level for publication in Nature. The work we present here is the first ever to image barium ions in the true working medium of a time projection chamber - that is, directly at the xenon gas-solid interface, in conditions that do not require ion extraction into an Paul trap or an ice layer in vacuum; we have also demonstrated a new and powerful form of microscopy: high-pressure gas-phase single molecule fluorescence imaging. The field of solid fluorescence sensors capable of single molecular imaging is being pioneered by the field of barium tagging but has significant implications to the structural design and understanding of catalytic processes, advanced energy storage materials, and molecular recognition and sensing in solid sensors for environmental and industrial purposes We assert that this major

step compares favorably to others that have been published in Nature journals in this area, and that it is of sufficiently wide interest (to microscopists, chemists, particle and nuclear physicists) and impact to justify publication in Nature Communications.

Minor comments

-Somewhere in the introduction it would be useful to explicitly mention the $0\nu\beta\beta$ reaction producing Ba^{2+} , e.g. $Xe-136 \rightarrow Ba-136^{2+} + 2e^-$

This has been fixed.

-The background index is mentioned as $b < 0.1(\text{ton yr ROI})^{-1}$, but this has wrong units (ct should be in the numerator), this should also be expressed in terms of exposure, i.e., ton-yr and ROI should be defined.

This has been fixed.

-FIG 1 and caption:

Minor spellings: "pressure chamber.r" \rightarrow "pressure chamber." **This has been fixed.**

The label "XYZ piezeo stages" \rightarrow "XYZ piezo stages." **This has been fixed.**

To improve readability, consider horizontal text or a vertical orientation of the entire figure.

We have rotated the text in this figure for improved readability.

The acronyms "LP" and "SP" should be fully written out in the figure or the caption to ensure that all readers clearly understand their meaning (it is only defined in the SI)

This has been fixed.

-The different patterns observed in the processed image shown in FIG. 3 are properly introduced, however the distinct features present in the left panel of FIG. 2—specifically, the straight lines with more dense bright points in two different parts of the image and the very intense blobs—are not addressed or discussed in either the text or the figure caption. Including an explanation or discussion of these features would enhance the reader's understanding and interpretation of the data presented.

This discussion has been added.

-The horizontal line in the right panel of FIG. 4 is not explained in the text or the figure caption. It is unclear what this line represents.

Added:

The horizontal line shows a cut that we place on this distribution for selecting single ion candidates spots.

And we have also added a new appendix on the analysis method, see below.

-The caption of FIG. 4 mentions that a single slide was used in the right figure, but this raises questions about the representativeness of the data. The manuscript should better describe how this slide was chosen. Was it selected based on certain criteria or randomly?

We have established the repeatability of these results over multiple slides (and in fact, for the air ones, across multiple microscopes). We pick one slide here to show the data in a presentable format. A comment has been added to the caption about the observed repeatability:

The plotted points are obtained over seven exposure regions on a single slide, and the trend is found to be repeatable over multiple slides. The horizontal line shows a cut that we place on this distribution for selecting single ion candidates spots.

-This point leads to a broader question: what criteria are used to determine that a bright spot is a signature of Ba-2+? The text mentions that "a small number of very weak emitters are present in the Ba-2+-free samples, though the bright spots associated with Ba-2+-bound IPG molecules are unambiguously identified as being present only in the Ba-2+-chelated sample." However, it is not clear how the distinction is made between Ba-2+-bound and Ba-2+-free samples. Is the differentiation based only on the intensity of the peak?

The statement, "A cut on the maximal identified step height in each trace is used to identify single molecule photo-bleaching event candidates," partly answers the question but it would be helpful to provide more details on this cut or even include a plot, to better illustrate the criteria used for identifying single molecule photo-bleaching events. This additional information would enhance the clarity and robustness of the presented data.

We have added a detailed new section of the Methods section that explicitly provides the mathematical manipulations applied to image sequences to draw these conclusions, as well as giving some examples of the intermediate stages.

In addition to providing these details, we identify that some of the confusion here may be due to our use of the possibly ambiguous word "samples", which could mean either samples in time, or sample preparations (the intended meaning in the places the reviewer identifies is the latter). We have replaced this word throughout wherever it refers to the barium chelated runs (ones where barium

was added) and unchelated runs (ones where barium was not added) with clearer language.

We also note that in order to report the algorithmic details as compactly as possible, we opted to simplify some of the mathematical image manipulation steps. As such some of the plots change a little, but none of the qualitative conclusions are affected.

-Right panel of Figure 6, the second plot (from the top) does not appear to represent a single barium ion chelated with IPG-1 as claimed. The presence of secondary peaks with large brightness suggests that multiple barium ions may be chelated rather than a single ion. Some clarification on this particular event would be beneficial to understand the observed data better and to confirm whether the fluorescence signal corresponds to a single barium ion or multiple ions.

This is correct. We have added:

“In the case of the second and fourth figures, nearby peaks from adjacent ions are also visible in the 3D image histograms.”

-In the sentence, "Such candidates are identifiable in all conditions tested, including ambient air, vacuum and and pressurized argon," remove 2nd "and"

This has been fixed.

-At the end of the first paragraph of Methods III.A has a period missing.

This has been fixed.

Reviewer #4 (Remarks to the Author):

Meaningful measurements are being made.

It would be worthy of publication if the minor corrections and responses below were made.

p2

- The unit should be corrected

$b < 0.1 \text{ (ct ton ROI)}^{-1}$  $b < 0.1 \text{ ct (yr ton ROI)}^{-1}$

This has been fixed.

p3

- wrong article number

[23] Journal of High Energy Physics 2018, 1 (2018)  Journal of High Energy Physics 2018, 112 (2018)

This has been fixed.

- What is the Ba^{2+} remaining efficiency ?

Even if recombination is low, it cannot be said that 100% of Ba^{2+} will survive.

There is a possibility that the charge is distributed by collisions with xenon atoms or impurities.

Are there any measured values or literature values for the Ba^{2+} survival efficiency?

This is (in our view) an extremely interesting question.

There are not measurements of this quantity directly, because it is very challenging to introduce test beams of Ba^{2+} into dense media- though experimental activity on this difficult problem is being pursued in both NEXT and nEXO collaborations. However, similar experiments with radon daughters show very high survival probability of the positive ion charge state following beta decays in xenon gas, and this extrapolates well to our situation of interest:

J. High Energ. Phys. 2018, 112 (2018).

The lack of recombination at pressures below 50 bar is also evident in energy resolution measurements, where recombination ultimately limits the performance of xenon drift chambers when the gas becomes very dense:

NIMA Volume 396, Issue 3, 11 September 1997, Pages 360-370

Providing a second, independent piece of evidence that recombination and consequent neutralization of Ba^{2+} following beta decay ought to be negligible.

It should be noted that in liquid, recombination of free electrons leads to further neutralization of the ion, and hence a distribution over charge states of Ba is expected in liquid xenon barium tagging experiments:

Phys. Rev. C 92, 045504

Both energetics considerations and the available data in similar but not identical conditions suggests that Ba^{2+} survival efficiency is expected to be very high and this guides our system design. To fully prove this will require a barium tagging

demonstration system with two neutrino double beta decays, which is under development as a long-term goal of the NEXT collaboration, though it lies outside the scope of the current paper.

p4

- Typo in FIG.1 caption
chamber.r  chamber.

Fixed.

- Explain abbreviation if you use in the latter text
Supplementary Information  Supplementary Information (SI)
done.

p5

- unit is not italic for "33 um"

Fixed

- In Fig.2, specify why lattice pattern inefficient region exist and it is affect to the Ba²⁺ tag measurement or not.

This has been clarified in the text. Given the way the images are stitched together, we expect the Ba²⁺ spots to be bright enough to be resolved everywhere in the relevant grids.

- Match the expression in the text in Fig2. caption
single molecule fluorescence  single molecule fluorescence candidate

Fixed,

- Put the condition of the measurement of Fig.2.

Do you put Ba²⁺ to the coverslip uniformly?

The Ba²⁺ is spin coated onto the cover slip as a salt mixed in with the chemosensor in these experiments, as discussed in the methods section. With this method, we do expect a fairly uniform coverage of the slide, though there may be some spatial dependence based on the way the solvent spreads under spinning (as evidenced by the streaks in the Ba²⁺ chelated images).

- What is the time for 1 step scan for 1mm² area?

We have added: “Since the exposure at each raster point 500~ms, a scan of this size is performed in approximately ten minutes”

p6

- In Fig.3 (right), some bright clusters seem to cut out at the edge of the 33um boxel. Why does it comes from?

We are not sure which voxel the reviewer means, but the bright spots in the image are associated with areas where more molecules were deposited, presumably as one of the droplets of solution evaporated after spin coating.

- In Fig.4 (right), specify black horizontal line.

Added:

The horizontal line shows a cut that we place on this distribution for selecting single ion candidates spots.

- In Fig.4 (right), why there is no event in the range of 0~3 and 46~50 of Step Time?

This is because the steps are identified by comparing activity before and after; there is no way to identify a step without some window of time before and after. This is clarified in a new appendix that explains the step-finding methodology in detail.

- What is the reason of existence of weak emitters in the IPG only sample?

We suspect these are dim fluorescence associated with unbound IPG molecules interacting with the glass surface. They are below the cut which we apply to identify single molecule candidates so can be considered as a harmless background, for present purposes.

p7

- In the text, Fig.6 is mentioned before Fig.5.

If there is no reason, it is better to swap them.

Done

- In Fig.5, why 1st step bin count is different in 10barXe and Air?

Is it just due to the amount of coating?

It is correct that not every slide is exactly equivalently prepared; also some candidates will photo bleach before the time that our algorithm can identify the

spot. But also, with a slower photobleaching lifetime, it is an expectation that less will bleach within the first step. These are histograms of the photobleaching step time, rather than the survival fraction. If we consider instead the surviving fraction histograms, it is clear that the total number of found spots is similar between both samples, but it the ones in the air sample bleach more quickly:

- What is the considered reason of the photo-blinking?

Photo-blinking is a typical feature in single molecule fluorescence imaging, understood to arise when molecules enter metastable “dark” conformations from which they cannot fluoresce; after some time they come back to the ground state and are active again. This is observed both in single-atom and single-molecule fluorescence experiments, though it is more common with molecules, since they have considerably more freedom to explore complex configurations.

p8

- Mean life times are inconsistent to that are written in Fig.5

Fixed. The earlier version had a typographical error. We have also updated the analysis a little since v1 to better handle the trajectories with multiple steps, and to provide a simpler prescription to describe in the new appendix - as such the values here change a little, but the conclusions do not. We add a comment on the method in the text:

Using the algorithm outlined in the Methods Section, which identifies the first significant (5σ in step confidence) transition in selected each time sequence, the photo-bleaching lifetime in 10-bar xenon gas is extracted to be 70.6~s, whereas in air it is much faster at 15.9~s, as shown in Fig.~\ref{fig:BaTimeConsts}. These values are extracted from accumulated barium ion candidates identified over seven long timescale runs of 500~s each.

And on the non-uniqueness of the method in the appendix:

Since some spots photo-blink in addition to photo-bleaching they may experience multiple transitions in one image sequence; as such these variables are not a unique or complete set of criterion by which to measure fluorescence activity, and more detailed quantification of images sequences $I_{i}(x,y)$ and brightness traces $B_{s}(x,y)$ is surely possible. Nevertheless, we have found these methods to be functionally useful for identifying single ion candidates and for quantitatively comparing fluorescence response in different conditions, the primary goals of the present work.

- What is the 15GB of data size correspond to? 1shot of the 1mm² image?
Reader cannot judge 15GB is large or not without any reference.

Correct. Clarified in text:

“15~GB uncompressed, for the single rastered image in Fig.~\ref{fig:Large-scale-image}”

p14

- typo: have have  have
This has been fixed.

We thank the reviewers for these final comments, and their excellent reviews that have improved the manuscript - we respond below.

Reviewer #1 (Remarks to the Author):

The authors have addressed all my concerns. I am particularly pleased with the new introduction, which clearly addresses aspects of Ba tagging that I have long wondered about while reading previous publications from the NEXT collaboration. Below, I list a few very minor comments on the introduction for the authors' consideration. However, I am already able to recommend the acceptance of this article in Nature and do not require reviewing the revised version.

Minor comments:

- Reference 7 is more than 20 years old and outdated. There are far more recent reviews available.

We have added Ref: Ann Rev Nucl Part Sci, 69, 219--251 (2019) for a more modern reference here.

- The paper by Schechter and Valle has some limitations. In particular, the diagram proposed in that work would generate only a very small Majorana mass (on the order of $1e-28$ eV), which is extremely so tiny compared to the experimentally estimated values from oscillation experiments. See, for instance, JHEP 06 (2011) 091.

We are aware of these complications - but it is challenging to review the field concisely in the introduction. To try to address this comment though, we tweak to “would guarantee generation of *at least a small* Majorana neutrino mass through loop corrections”

Double-beta decay specifically refers to neutrons converting to protons and leptons, not protons converting to neutrons. Should the authors want to mention it, double-electron capture is the other process that is actively experimentally investigated.

We understand that the name “neutrinoless double beta decay” can also be applied to processes where two protons turn to two neutrons and two positrons - these are less common on account of the proton to neutron mass difference, but are postulated to occur in some isotopes. See for example:

<https://www.sciencedirect.com/science/article/abs/pii/0146641084900061>

About electron capture, while it is experimentally studied (including with NEXT), the limits are currently far from the sensitivity of $0\nu\beta\beta$ when quantified via the neutrino mass reach. Because of space constraints we prefer not to add discussion of this here - reviewer indicates that this is ok.

- The sentence "likely the only observed manifestation of physics above the electroweak scale" is not trivial, and I am unsure whether it is correct. Please provide a reference to support this claim.

We have replaced “likely the only observed” with “potentially the only directly observed” to soften this claim a little, since we agree that reasonable people could disagree about both what constitutes an observation and what constitutes of physics above EW scale.

- Please include the confidence level of the half-life limit for Xe.

Added “at 90% confidence level”

- The rate of $0\nu\beta\beta$ decay depends on several other factors, particularly the Majorana phases, which are currently the primary source of uncertainty along with the lightest neutrino mass eigenstate. Nuclear matrix elements and the phase space factor may also be worth mentioning.

Added: “The expected rate of $0\nu\beta\beta$ depends on the neutrino masses and mixing parameters, nuclear matrix elements~\cite{engel2017status}, and phase space factors~\cite{stoica2019phase}.”

- In the sentence "loose spatio-temporal requirement," would it be possible to quantify the spatial resolution? This is something I have always wondered about but have never seen addressed. Is there any specific work that quantifies this requirement that could be cited?

The rate of ordinary double beta decay is one per 5 kg per day. The density of xenon gas at 10 bar is 57 kg m⁻³ so there are roughly ten background barium ions per m³ per day. The coincidence needs to be constraining enough to confidently reject this background.

There is a tradeoff between time and space resolution needed to achieve this. As an example, if the origin of the barium ion can be established to a cube with 10cm on a side, or 1e-3 m³, the expected Ba²⁺ production in this region is 0.1 per day. To have 99.9% confidence in this barium ion not originating from a background event, a coincidence should be established to within 1% of a day, or 900 seconds.

This sort of condition seems fairly straightforward to meet, but because the exact scheme to provides a specific space or time resolution is not firm yet, it seems better not to go into these kinds of details in the manuscript.

- The process of Ba capture and subsequent detection is complex, and the authors rightly state that the compounded efficiency requirements are high. They point out some work discussing capture efficiencies, but perhaps the addition of a small comment focusing on the imaging/detection efficiency, as this is the topic of the article, will help convince the reader that such efficiency requirements might be achievable with the proposed methods.

The following comment has been added:

Given the large observed binding constants of the crown-ether-based dyes~\cite{miller2023barium}, this mode of microscopy appears well suited to providing an efficient barium imaging technique for future xenon ^{135}Xe experiments.

I would suggest proofreading the manuscript, as I noticed a few typos. For example, in the conclusions, “A rational zero suppression algorithm and a form of online trigger for frames of interest will be advantageous...”

We have fixed this sentence and reviewed the rest of the article.

Reviewer #2 (Remarks to the Author):

Reviewer #3 (Remarks to the Author):

We thank the authors for thoroughly addressing our comments and are satisfied with all answers. The manuscript has become more approachable to a wider audience and we think that the authors have provided convincing arguments for publishing this paper in Nature Communications. We therefore recommend acceptance.

Reviewer #4 (Remarks to the Author):

The authors have made most of the necessary corrections and clarifications.

> We suspect these are dim fluorescence associated with unbound IPG molecules interacting with the glass surface. They are below the cut which we apply to identify single molecule candidates so can be considered as a harmless background, for present purposes. Some points in most right figure of Fig.4 seem to be over the cut threshold (1000). If the threshold were higher (like 4000), that explanation would make sense.

Thank you for this observation.

The cut is applied to select candidates for further analysis, and there always will be some small tail of background events that do pass it (it is always question of the desired signal efficiency vs background rejection power, in such event selections). As is seen from the figure, many more pass with much brighter emission in the with Ba^{2+} samples. The

position of this cut and the subsequent analysis applied to the waveforms will eventually depend on the implementation in a Ba²⁺ double beta decay experiment, to be further optimized in future - the cut at 1000 here is applied to select candidates for further analysis and does a good job of rejecting most of the dim background spots while selecting the bright ones that are present when Ba²⁺ is added.

There is very minor comment about the reference of [63] Thorlabs FEHH0500, I can not find the product with this model number. I wonder this is typo of FESH0500.

Thank you for catching this typo. It is now fixed.